# `G1`: Teaching LLMs to Reason on Graphs with Reinforcement Learning

**Xiaojun Guo**[1*]    **Ang Li**[1*]    **Yifei Wang**[3*]
**Stefanie Jegelka**[4,3]    **Yisen Wang**[1,2†]

[1]State Key Lab of General Artificial Intelligence,
School of Intelligence Science and Technology, Peking University
[2]Institute for Artificial Intelligence, Peking University
[3]MIT
[4]TUM

## Abstract

Although Large Language Models (LLMs) have demonstrated remarkable progress, their proficiency in graph-related tasks remains notably limited, hindering the development of truly general-purpose models. Previous attempts, including pretraining graph foundation models or employing supervised fine-tuning, often face challenges such as the scarcity of large-scale, universally represented graph data. We introduce `G1`, a simple yet effective approach demonstrating that Reinforcement Learning (RL) on synthetic graph-theoretic tasks can significantly scale LLMs' graph reasoning abilities. To enable RL training, we curate `Erdős`, the largest graph reasoning dataset to date, comprising 50 diverse graph-theoretic tasks of varying difficulty levels, 100k training data and 5k test data, all drived from real-world graphs. With RL on `Erdős`, `G1` obtains substantial improvements in graph reasoning, where our finetuned 3B model even outperforms Qwen2.5-72B-Instruct (24x size). RL-trained models also show strong zero-shot generalization to unseen tasks, domains, and graph encoding schemes, including other graph-theoretic benchmarks as well as real-world node classification and link prediction tasks, without compromising general reasoning abilities. Our findings offer an efficient, scalable path for building strong graph reasoners by finetuning LLMs with RL on graph-theoretic tasks, which combines the strengths of pretrained LLM capabilities with abundant, automatically generated synthetic data, suggesting that LLMs possess graph understanding abilities that RL can elicit successfully. Our implementation is open-sourced at `https://github.com/PKU-ML/G1`, with models and datasets hosted on Hugging Face collections PKU-ML/G1 for broader accessibility.

## 1   Introduction

Large Language Models (LLMs) have achieved widespread success [2, 13] but exhibit notable limitations in reasoning about graph-structured data, a critical capability for achieving general-purpose intelligence. Proficient graph reasoning is essential for numerous applications, yet even state-of-the-art LLMs like OpenAI's o1 [30] demonstrate significant deficiencies, with reported accuracies as low as 58.49% on graph connectivity tests [55].

Initial efforts to enhance LLMs' graph understanding explored various natural language encoding schemes [11, 5, 9], but these yielded only modest improvements. Alternative strategies have involved

---

*Equal Contribution.
†Corresponding Author: Yisen Wang (yisen.wang@pku.edu.cn).

39th Conference on Neural Information Processing Systems (NeurIPS 2025).

Table 1: An overview of 50 graph-theoretic tasks in our dataset `Erdős` (100k train, 5k test), alongside with the difficulty distribution, and the accuracy of the base model Qwen2.5-7B-Instruct and our RL-trained `G1-7B` model. A complete description of tasks are in Appendix K.2.

| Difficulty | Tasks | Ratio | Base Model Acc | `G1` Acc |
|---|---|---|---|---|
| **Easy** | Node Number, Dominating Set, Common Neighbor, Edge Number, Neighbor, BFS, Has Cycle, DFS, Minimum Spanning Tree, Edge Existence, Is Regular, Degree, Is Tournament, Density | 29.16% | 57.16% | **95.07%** |
| **Medium** | Adamic Adar Index, Clustering Coefficient, Connected Component Number, Bipartite Maximum Matching, Local Connectivity, Jaccard Coefficient, Min Edge Covering, Is Eularian, Degree Centrality, Is Bipartite, Resource Allocation Index | 22.91% | 42.55% | **88.91%** |
| **Hard** | Max Weight Matching, Closeness Centrality, Traveling Salesman Problem, Strongly Connected Number, Shortest Path, Center, Diameter, Barycenter, Radius, Topological Sort, Periphery, Betweenness Centrality, Triangles, Average Neighbor Degree, Harmonic Centrality, Bridges | 33.33% | 18.87% | **50.44%** |
| **Challenging** | Isomophic Mapping, Global Efficiency, Maximal Independent Set, Maximum Flow, Wiener Index, Hamiltonian Path, Min Vertex Cover | 14.58% | 3.29% | **23.57%** |

instruction tuning [27, 53] or preference tuning [3, 42] on curated graph datasets. Others attempted to build specialized graph foundation models through pretraining [28, 21, 26]; however, these are often limited by the lack of large-scale, universal graph representations suitable for diverse graph types. Different from prior work, we believe LLMs pretrained on Internet-scale data already possess graph reasoning ability, and we can elicit it through their own trial and error without human data.

In this work, we are the first to explore the use of Reinforcement Learning (RL) to solve graph reasoning tasks. We chose graph-theoretic problems as a testbed as they allow direct verification of generated answers to produce rule-based rewards for RL training, which is shown to be key for the success of DeepSeek R1 in math and coding problems [13]. We collect the largest-to-date graph-theoretic problem set, `Erdős`, with either groundtruth answers or automatic verification programs. As illustrated in Table 1, these tasks span a wide spectrum of difficulty levels, from basic graph properties like node counting to NP-hard problems such as finding the maximal independent set. Another advantage of adopting graph-theoretic tasks is its circumvention of scarce human-annotated data; the model learns through exploration and reinforcement on synthetic tasks where ground-truth outcomes provide direct reward signals, similar to the AlphaGo-Zero paradigm [37]. Besides data construction, we also study various aspects of the training process, such as the influence of data mixture, supervised initialization, and the use of chain-of-thought [45]. Our results confirm that RL with synthetic graph-theoretic task is a powerful and scalable approach to improving graph reasoning abilities of LLMs.

Our work makes the following key contributions:

- We are the first to apply reinforcement learning (RL) framework to improving LLMs on graph reasoning tasks. The resulting model `G1` significantly enhances the graph reasoning abilities of LLMs across a diverse set of synthetic tasks, demonstrating that appropriately finetuned LLMs can become stronger graph reasoners.

- We introduce `Erdős`, the largest-scale and most comprehensive graph-theoretic dataset that comprises 50 distinct tasks of varying complexities, uniquely constructed from diverse real-world graphs, providing a reliable platform for training and evaluating graph reasoning.

- We empirically demonstrate that `G1` achieves substantial performance improvements on our `Erdős` benchmark, with gains of up to 46% over baseline models. Notably, our fintuned `G1-7B` model attains competitive performance with state-of-the-art reasoning models like OpenAI's o3-mini and `G1-3B` easily rivals Qwen2.5-72B-Instruct by noticeable margins.

- `G1` models exhibit strong zero-shot generalization on unseen graph tasks and domains, improving base models' performance on other graph-theoretic benchmarks (GraphWiz and GraphArena) and real-world graphs (Cora and PubMed) without deteriorating general reasoning ability (GSM8K, MATH, and MMLU-pro), indicating a synergetic improvement of LLMs' graph reasoning abilities through RL.

`G1` charts a data-efficient and scalable course for developing LLMs with strong graph reasoning. By demonstrating that RL can unlock latent graph understanding within general-purpose LLMs using synthetic data, our work suggests a possible paradigm shift away from reliance on heterogeneous real-world graphs to build graph foundation models. This paves the way for more versatile AI systems capable of sophisticated reasoning across diverse data modalities.

## 2  Related Work

**Graph Reasoning.** Graph reasoning problems fall into two categories: domain-specific, which require understanding both graph structures and node/link attributes, *e.g.*, node classification, link prediction, and knowledge-based QA [15, 56, 18]; and domain-agnostic, also called *graph-theoretic problems*, which focus solely on structural reasoning but find a lot of practical uses in various domains, *e.g.*, shortest paths, Hamiltonian paths, graph isomorphism [50, 35]. For the latter problems that we study in this paper, people have studied the use of RL [29, 40] or unsupervised learning [19], often in conjunction with Graph Neural Networks (GNNs) [20, 49] that align with the solution structure [51]. Yet these models are often built to solve each problem alone. Recently, Sanford et al. [34] prove and validate the priority of the transformer models compared to GNNs on complex graph reasoning tasks requiring long-range dependencies. In this work, we focus on building general-purpose graph reasoners that could solve a range of graph-theoretic problems by exploiting the strength of LLM pretraining, and find that the ability also generalizes to the former domain-specific graph tasks.

**Benchmarking LLMs on Graph Reasoning.** There is a growing interest in evaluating LLMs' graph reasoning abilities. NLGraph [41] evaluate LLMs on graph-theoretic tasks and discover preliminary yet brittle reasoning abilities in the face of spurious correlations and large graphs. Later, GraphArena [38] and GraCoRe [55] include a broader task coverage and recently released LLMs, finding that even OpenAI o1-mini struggles a lot with complex tasks. Moreover, GraphEval2000 [46] and ProGraph [23] emphasize code-oriented problem solving using library-based prompts, and GraphOmni [48] unify varying graph types, encodings, and prompt styles for a comprehensive evaluation. Overall, these benchmarks suggest that LLMs overall demonstrate moderate success on simple tasks but struggle with abstraction, generalization, and larger or more complex graph instances. Nevertheless, these datasets are either too small (e.g., thousands of examples) or not diverse enough (e.g., 8 tasks in NLGraph) for training general-purpose graph reasoners, which motivates the design of `Erdős`.

**Improving LLMs on Graph Reasoning.** A major concern when using LLMs for graph tasks is the mismatch of data structure: LLMs take text sequences as input, while graphs have no natural order. Fatemi et al. [11] analyzed different graph encoding schemes for LLMs, such as adjacency lists and real-name networks, revealing that no single strategy proved universally optimal across all tasks and models. Subsequent explorations with different linearization orders [5], graph embeddings [31], or input modalities [9] have generally resulted in only modest improvements. Another thread of research proposes post-training LLMs using instruction tuning [27, 53] or preference tuning [3, 42, 39] on curated datasets of graph problems. However, the creation of diverse, high-quality instruction datasets at scale is challenging and expensive and requires extra supervision. Furthermore, models trained via distillation may only learn to memorize patterns and overfit to graph tasks [4]; in Section 5.2, we show that previous instruction-tuned models exhibit dramatic failures when generalizing to other data formats and reasoning tasks, while our RL training yields consistently better performance.

**Reinforcement Learning for LLMs Reasoning.** Recent advances have demonstrated that LLMs can attain strong reasoning abilities in math and coding domains through RL, with representative work like OpenAI o1 [30] and DeepSeek R1 [13]. However, as discussed above, even o1 struggles a lot with graph reasoning tasks [55] and it is thus yet unclear whether RL can reliably and scalably improve LLMs' graph reasoning abilities. Our findings on `G1` first confirm the effectiveness of RL on graph reasoning as well and suggest that applying RL to diverse graph-theoretic tasks with verifiable rewards is a scalable path for eliciting generalizable graph reasoning abilities of LLMs.

## 3  `Erdős`: A Comprehensive Collection of Graph-theoretic Reasoning Tasks on Real-world Graphs

To facilitate rule-based Reinforcement Learning of LLMs (aka. Reinforcement Learning from Verifiable Rewards (RLVR)) on graphs, we construct a diverse, large-scale collection of graph-theoretic reasoning tasks. We name it `Erdős` to remember Paul Erdős, a seminal figure with diverse contribu-

tions to graph theory. Compared to real-world graph tasks, these graph-theoretic tasks allow clear rule-based determination of rewards for the answers sampled from LLMs. We categorize these tasks into **Easy, Medium, Hard, and Challenging**, based on their inherent problem complexity as well as current LLMs' ability to solve them (see a full list in Table 1). For the training split, there are a total of 100,000 question-answer pairs, evenly distributed across tasks with 2,000 examples each. We also reserve 5,000 test pairs with different questions for evaluation. We include a detailed comparison of Erdős with other graph reasoning benchmarks in Appendix K.1. Erdős can serve as a dataset for training LLMs as well as a benchmark for evaluating LLMs on graph-theoretic tasks. We will release all task prompts, problems, chain-of-thought exemplars, and solution verification programs for public use. Below is a more detailed description of the data collection process.

**Graph-theoretic Tasks.** We curate 50 graph-theoretic reasoning tasks available on NetworkX [14], one of the most widely used library for graph processing, and construct, as we know, the most comprehensive collection so far. In the difficulty level, the tasks vary from easy determination of graph attributes like node number counting, to well-known NP-hard problems like the traveling salesman problem. This collection includes both tasks for general graphs and tasks specific to directed graphs or weighted graphs, and covers a wide range of answer types including boolean, integer, float, node list, edge list, and node mapping.

**Answer Generation.** To generate the golden answer for each problem, we utilize the default solvers of NetworkX to automatically solve the problem. If there are multiple solutions to each question, we use NetworkX-based programs to verify the correctness of each generated solution. The procedure ensures rigorous rewarding attribution, avoiding both costly human labeling and potential bias and hacking brought by LLM judges.

**Graph Sources.** Most previous graph-theoretic datasets or benchmarks [41, 27, 3] consider random graphs, following Erdős-Rényi model [10] or Barabási–Albert model [1]. However, these random graph models are often far from graphs encountered in real-world practice. To mitigate this gap, we utilize the real-world graphs from the Network Repository [33], the largest network repository with thousands of donations in 30+ domains. As these graphs can be very large and infeasible for LLMs, we downsample the graphs by random walk with a restart strategy, generating subgraphs with sizes from 5 to 35 nodes, following common settings in previous work [41, 55, 38].

**Language Encoding**. There are multiple ways to translate the graph structure into languages that LLMs can understand. Previous works explore serialized formats such as adjacency matrix, edge list, or graph embeddings [11, 8, 53], but fail to find a consistently good method. Here, we choose to describe the graph structure in a unified edge list format, *e.g.*, $(1, 2), (2, 3), \ldots$. In later experiments of Section 5.2, we show that our model trained on a single graph description method can even positively transfer to other formats.

# 4 Training LLMs to Reason on Graphs

In this section, we introduce the training pipeline that we explored for training G1. We design proper rule-based rewards for different graph tasks, while intentionally keeping the RL algorithm general and consistent with previous work. Similar to DeepSeek R1 [13], the training of G1 is very simple: it consists of a Reinforcement Learning phase for rewarding correct rollouts with the GRPO algorithm [36], and an *optional* SFT phase for warming up the model in the beginning (without which we call G1-Zero). We find that the SFT phase is generally beneficial for learning more challenging tasks, whose initial accuracy with the base model is close to zero.

## 4.1 Reinforcement Learning of LLMs on Graphs

**Rule-based Rewards on Graphs.** We design the following rule-based outcome reward model (ORM) for our training on graph-theoretic tasks, with a combination of value match, set matching, and algorithmic verification for different problems:

- *Strict value matching*. For tasks that have a unique ground truth value, e.g., node counting, the policy receives a reward of +1 only when the generated answer is identical to the ground truth in terms of numerical value, e.g., 0.5 and 1/2, otherwise it receives a reward of 0.
- *Jaccard Index for set matching.* For problems whose answer is not a single value $\hat{s}$ but an unordered set, e.g., common neighbors of two nodes, the reward is defined as the Jaccard

Index between the generated set $\hat{s}$ and the ground truth $s$, i.e., $|s \cap \hat{s}|/|s \cup \hat{s}|$. In this way, the model can receive intermediate rewards for imperfect solutions.

- *Algorithmic verification.* Lastly, for problems that have multiple correct solutions (e.g., shortest paths) and it is not feasible to enumerate all of them, we implement algorithmic verifiers to check correctness of the proposed solutions. For instance, we determine the validness of a Hamiltonian path proposed by the policy by checking whether all the edges in the path exist and each node is visited exactly once.

**RL Algorithm.** Following common practice [13], we use the Group Relative Policy Optimization (GRPO) [36] algorithm for RL training. Specifically, for each question $q \sim P(Q)$ drawn from the training set, GRPO first samples a set of responses $\{o_i\}_{i=1}^{G}$ from the policy model. The responses receive rewards $\{r_i\}_{i=1}^{G}$, which enables calculating the group relative advantages $\{A_i\}_{i=1}^{G}$:

$$A_i = \frac{r_i - \mathrm{mean}(\{r_1, r_2, \cdots, r_G\})}{\mathrm{std}(\{r_1, r_2, \cdots, r_G\})}. \tag{1}$$

Next, the policy model $\pi_\theta$ is updated by maximizing the following objective:

$$\mathcal{J}_{\mathrm{GRPO}}(\theta) =$$

$$\mathbb{E}_{q, \{o_i\}_{i=1}^{G}} \frac{1}{G} \sum_{i=1}^{G} \left( \min \left( \frac{\pi_\theta(o_i|q)}{\pi_{\theta_{\mathrm{old}}}(o_i|q)} A_i, \mathrm{clip}\left( \frac{\pi_\theta(o_i|q)}{\pi_{\theta_{\mathrm{old}}}(o_i|q)}, 1 - \epsilon, 1 + \epsilon \right) A_i \right) - \beta \mathbb{D}_{KL}\left( \pi_\theta || \pi_{\mathrm{ref}} \right) \right), \tag{2}$$

where the expectation is taken over $q \sim P(Q)$ and $\{o_i\}_{i=1}^{G} \sim \pi_{\theta_{old}}(O|q)$. The KL divergence to the reference policy $\pi_{\mathrm{ref}}$ (base model) prevents large deviation from the pretrained model and circumvents severe overfitting. Besides, $\epsilon$ controls the clipping range of the probability ratios.

### 4.2 Optional Warm-up with Supervised Fine-tuning

During RL training, we have noticed that for some challenging tasks like isomorphic mapping (see Table 1), the initial accuracy of the base model is often so low that we frequently end up with only incorrect rollouts, producing no useful signal for RL training. This issue can be mitigated by using a stronger base model with higher initial performance; for example, R1 uses DeepSeek V3 (671B parameters) as its base model, although this inevitably increases compute cost. We find that introducing a short warm-up phase with supervised fine-tuning, aimed at teaching the model basic reasoning skills before the RL phase, effectively improves overall learning efficiency. Specifically, in this paper we consider two types of supervised fine-tuning.

**Direct-SFT.** The first is direct supervised fine-tuning on question-answer pairs $(q, a)$, where $q$ is the textual description of the problem and $a$ is the final answer without any intermediate reasoning steps. As discussed above, for graph-theoretic tasks, these question-answer pairs can often be synthesized by programming. However, this approach does not include the reasoning steps leading to the answers, meaning we cannot use it to explicitly teach the model reasoning processes.

**CoT-SFT.** Secondly, we can collect reasoning trajectories via sampling $(q, c, a)$ triplets from another model [54], where $c$ represents the Chain-of-Thought (CoT) reasoning steps in natural language that lead to the final answer $a$, and use them to fine-tune the base model. Specifically, we instruct a base model to generate potential solutions for each question $q$, and only keep the correct responses that pass verification. This process is also called Rejection Sampling Fine-tuning (RFT) [54]. In practice, we use Qwen2.5-32B-Instruct [32], a more capable model for generating candidate solutions more reliably, ending up with around 4,500 training examples for the SFT phase.

## 5 Experiments

### 5.1 Benchmarking `G1` on Graph-theoretic Reasoning Tasks

**Setup.** As shown in Table 2, in the interest of academic compute budgets, we focus on comparing relatively small models. We include strong proprietary models (of unknown sizes) like GPT-4o-mini (non-reasoning) and OpenAI o3-mini (state-of-the-art reasoning), open-source instruction models like Qwen2.5-Instruct series (3B, 7B, 72B) [32], Qwen2.5-Math-Instruct [52], LLaMA-3 series (3B, 8B,

Table 2: Test accuracy (%) comparison of different LLMs of varying sizes on our Erdős benchmark tasks. In all experiments we use Qwen2.5-Instruct models as our base model (marked below). We report the average accuracy across all tasks in the *Average* column, and full results for each task are provided in Appendix I.4.

| Model | Easy | Medium | Hard | Challenging | Average |
|---|---|---|---|---|---|
| Proprietary (Unknown Parameters) | | | | | |
| GPT-4o-mini | 76.20 | 72.07 | 28.81 | 3.34 | 47.60 |
| OpenAI o3-mini (w/ tool use) | 74.83 | 83.49 | 59.28 | 43.22 | 64.90 |
| 3B Parameters | | | | | |
| Llama-3.2-3B-Instruct | 36.50 | 21.45 | 6.81 | 1.14 | 17.32 |
| Qwen2.5-3B-Instruct (base model) | 45.71 | 30.18 | 9.44 | 1.29 | 22.72 |
| Direct-SFT-3B (Ours) | 74.43 | 75.27 | **43.69** | **14.43** | 53.78 |
| CoT-SFT-3B (Ours) | 65.57 | 67.64 | 29.44 | 4.57 | 43.56 |
| **G1-3B (Ours)** | **94.86** | **84.64** | 41.25 | 7.57 | **59.76** (+37.04) |
| 7B Parameters | | | | | |
| Llama-3.1-8B-Instruct | 49.21 | 30.45 | 13.69 | 1.43 | 25.10 |
| Qwen2.5-7B-Instruct (base model) | 57.36 | 42.55 | 18.87 | 3.29 | 32.06 |
| Qwen2.5-Math-7B-Instruct | 52.79 | 39.64 | 14.82 | 2.46 | 28.94 |
| DeepSeek-R1-Distill-Qwen-7B | 71.79 | 73.73 | 39.12 | 16.57 | 51.64 |
| GraphWiz-7B-RFT | 14.57 | 13.73 | 1.38 | 0.47 | 7.70 |
| GraphWiz-7B-DPO | 20.36 | 19.09 | 1.44 | 0.78 | 10.59 |
| Direct-SFT-7B (Ours) | 73.57 | 75.91 | 39.12 | 10.71 | 51.76 |
| CoT-SFT-7B (Ours) | 72.57 | 75.73 | 38.50 | 11.00 | 51.34 |
| **G1-7B (Ours)** | **95.07** | **88.91** | **50.44** | **23.57** | **66.16** (+34.10) |
| 70B Parameters | | | | | |
| Llama-3.1-70B-Instruct | 68.07 | 55.45 | 31.87 | 4.44 | 42.28 |
| Qwen2.5-72B-Instruct | 71.71 | 67.81 | 33.37 | 8.22 | 47.16 |

70B) [12], and a strong baseline DeepSeek-R1-Distill-Qwen-7B [13] that is distilled from DeepSeek R1 with 671B parameters. Additionally, for reference, we incorporate previous training strategies for graph reasoning tasks such as GraphWiz-RFT and GraphWiz-DPO [3]. We finetune our model from Qwen2.5-Instruct models (3B and 7B) for 300 steps with batch size 512 on a cluster of $8\times$A800 GPUs, using our dataset Erdős. More experimental details can be found in Appendix A.

**Performance.** As shown in Table 2, our proposed model G1-7B consistently outperforms most proprietary, open-source, and graph training counterparts by significant margins across all difficulty levels. With a notable average accuracy of 66.16%, G1-7B outperforms GPT-4o-mini (47.60%) by 18.56%, reaching competitive performance to a cutting-edge reasoning model like o3-mini (64.90%) that underwent much heavier training. Notably, our small variant G1-3B, delivers a strong average performance of 59.76%, surpassing open-source models including Qwen2.5-72B-Instruct (47.16%) and Llama-3.1-70B-Instruct (42.28%) with $20\times$ parameters. We also evaluate the GPT-4o model on a randomly sampled subset of Erdős due to the cost budget. As shown in Appendix Table 14, G1-7B surpasses GPT-4o across all difficulty levels, exceeding it by over 10% on average (65.29% vs. 55.13%), further validating the strong graph reasoning capabilities of the G1 models.

**Remark on SFT baselines.** Interestingly, Direct-SFT emerges as a surprisingly strong baseline in Table 2. The 3B and 7B versions of Direct-SFT both outperform larger open-source models with 53.78% and 51.76% accuracy, suggesting that LLMs can discover some effective patterns by directly fitting targets. However, we also observe that with Direct-SFT, the 7B model yields no extra gain over the 3B model, while CoT-SFT and G1 (initialized with CoT-SFT) performance scales with larger models. This indicates that even though the CoT-SFT performance may appear low compared to Direct-SFT (possibly because of limited data size with about 100 examples per task), CoT-SFT could have better scaling and generalization properties.

**Robustness Analysis.** To rigorously evaluate robustness, we conducted 32 repeated runs with different random seeds. The results in Appendix Table 17 demonstrate consistently small standard deviations (<1% across all models and difficulty levels), confirming the stability of our method against potential randomness in LLM outputs. For prompt robustness, we rigorously test prompt sensitivity by having GPT-4o generate three semantically equivalent prompt variants. Appendix Table 18 shows minimal performance variance (<1.5% standard deviation) across all models and difficulty levels, confirming our benchmark's stability to phrasing changes.

**Scaling `G1` to 32B.** To demonstrate the scalability of our approach, we extended the training methodology to develop `G1`-Zero-32B from Qwen2.5-32B-Instruct. As shown in Table 3, `G1`-Zero-32B achieves a **27.96%** improvement in average accuracy (from 47.10% to 75.06%), with particularly notable gains in harder categories: +31.87% on Hard problems and +26.43% on Challenging problems. Furthermore, Appendix Table 16 demonstrates that `G1`-Zero-32B not only preserves but slightly enhances mathematical performance on standard benchmarks, which shows modest improvements on both GSM8K (+0.08%) and MATH (+4.00%).

Table 3: Test accuracy (%) of `G1`-Zero-32B and Qwen2.5-32B-Instruct on `Erdős`.

|  | Easy | Medium | Hard | Challenging | Average |
|---|---|---|---|---|---|
| Qwen2.5-32B-Instruct | 70.57 | 68.73 | 33.38 | 9.00 | 47.10 |
| **G1-Zero-32B** | **97.79** | **93.00** | **65.25** | **35.43** | **75.06** |

**Scaling `G1` to larger graphs.** To verify the transferability of `G1` to larger graphs, we construct a new test set of 5,000 graphs with 36-100 nodes, with other settings kept the same, which ensures there is no overlap between training and test data. Table 4 shows that both `G1`-3B and `G1`-7B achieve strong zero-shot generalization to these larger graphs without additional training, significantly outperforming the baselines across difficulty levels. These results demonstrate our method's effective scalability beyond the original training distribution. For larger graphs with thousands of nodes, `G1` is bottle-necked by the context window limit of underlying LLMs, detailed in Appendix D.

Table 4: Zero-shot generalization (accuracy in percentage) of `G1` to larger graphs with 36-100 nodes.

|  | Easy | Medium | Hard | Challenging | Average |
|---|---|---|---|---|---|
| Qwen2.5-3B-Instruct | 27.98 | 28.53 | 5.26 | 0.29 | 16.74 |
| **G1-3B** | **79.39** | **65.66** | **18.46** | **3.74** | **44.29** |
| Qwen2.5-7B-Instruct | 37.86 | 41.56 | 9.17 | 1.17 | 23.94 |
| **G1-7B** | **76.65** | **70.67** | **23.16** | **5.22** | **46.46** |

## 5.2 Transferability of `G1` to Unseen Tasks and Domains

In this section, we evaluate *zero-shot* generalization of `G1` to unseen domains, tasks, and data formats. Detailed benchmark description and complete evaluation setups are provided in Appendix B.

### 5.2.1 `G1`'s Transferability to Other Graph Reasoning Benchmarks

**Setup**. We consider two additional graph reasoning benchmarks, *GraphWiz* [3] and *GraphArena* [38], which bring three major shifts that challenge our model: 1) different distributions of the underlying graphs 2) tasks unseen during training 3) unfamiliar graph encoding formats, e.g., the GraphArena benchmark represents nodes with human names instead of integers.

**Results.** The performance across models is reported in Table 5 and Table 6. On the GraphWiz benchmark, `G1`-7B achieves the highest overall accuracy (57.11%) among all models, outperforming DeepSeek-R1-Distill-Qwen-7B (51.86%) and even models specifically trained on GraphWiz data such as GraphWiz-7B-RFT (49.61%). The smaller variant `G1`-3B also achieves comparable performance with DeepSeek-R1-Distill-Qwen-7B. Similar results can be found on the GraphArena benchmark (Table 6) with a different graph encoding scheme. These results demonstrate that `G1` has strong zero-shot generalization ability to unseen graph encoding methods, graph distributions, and graph tasks. Full results for GraphWiz and GraphArena are shown in Appendix I.1 and Appendix I.3.

### 5.2.2 G1 on Real-world, Non-graph-theoretic Graph-reasoning Tasks

**Baseline.** For real-world graph tasks, we consider two standard problems: node classification and link prediction. We adopt the benchmarks introduced by Wang et al. [44], which are constructed by subsampling from the widely used Cora and PubMed citation graphs. Each instance includes a description of the target node (or node pair) containing the paper ID and title, along with the textual

Table 5: Test accuracy (%) by computational complexity on the GraphWiz benchmark.

| Model | Linear | Poly | NP-Complete | Avg. |
|---|---|---|---|---|
| Llama-3.2-3B-Instruct | 29.80 | 3.00 | 2.50 | 19.80 |
| Qwen2.5-3B-Instruct (base) | 40.25 | 9.58 | **69.12** | 36.44 |
| **G1-3B** | **58.06** | **26.75** | **69.12** | **50.08** |
| Llama-3.1-8B-Instruct | 54.00 | 5.67 | 32.12 | 33.03 |
| DeepSeek-R1-Distill-Qwen-7B | 57.69 | 31.42 | 70.88 | 51.86 |
| GraphWiz-7B-RFT | 67.56 | 29.83 | 43.38 | 49.61 |
| GraphWiz-7B-DPO | 63.88 | **36.25** | 39.50 | 49.25 |
| Qwen2.5-7B-Instruct (base) | 49.06 | 17.92 | **76.12** | 44.69 |
| **G1-7B** | **68.00** | 32.25 | 72.62 | **57.11** |

Table 6: Test accuracy (%) by computational complexity on the GraphArena benchmark.

| Model | Poly-Time | | NP-Complete | | Avg. |
|---|---|---|---|---|---|
| | Easy | Hard | Easy | Hard | |
| Llama-3.2-3B-Instruct | 22.25 | 6.75 | 8.00 | 0.66 | 8.40 |
| Qwen2.5-3B-Instruct (base) | 31.50 | 14.50 | 17.33 | 1.50 | 14.85 |
| **G1-3B** | 57.50 | 26.75 | 24.66 | 1.83 | 24.80 |
| Llama-3.1-8B-Instruct | 47.00 | 21.25 | 22.00 | 2.16 | 20.90 |
| DeepSeek-R1-Distill-Qwen-7B | 66.0 | 22.75 | 34.83 | 1.50 | 28.65 |
| GraphWiz-7B-RFT | 2.25 | 0.75 | 0.83 | 0.00 | 0.85 |
| GraphWiz-7B-DPO | 0.25 | 1.00 | 0.66 | 0.16 | 0.49 |
| Qwen2.5-7B-Instruct (base) | 62.00 | 35.75 | 28.83 | 2.16 | 28.84 |
| **G1-7B** | **77.50** | **44.25** | **47.33** | **8.50** | **41.10** |

Table 7: Test accuracy (%) on Node Classification and Link Prediction benchmarks.

| Model | Node | | Link | | Avg. |
|---|---|---|---|---|---|
| | Cora | PubMed | Cora | PubMed | |
| Llama-3.2-3B-Instruct | 68.77 | 75.20 | 60.40 | 57.60 | 64.79 |
| Qwen2.5-3B-Instruct (base) | 70.83 | 75.08 | 62.15 | 58.38 | 65.66 |
| CoT-SFT-3B | 75.97 | 81.47 | 75.70 | **71.52** | 75.12 |
| **G1-3B** | **77.25** | **83.88** | **78.97** | 69.75 | **75.16** |
| Llama-3.1-8B-Instruct | 70.90 | 75.00 | 50.60 | 46.10 | 59.53 |
| DeepSeek-R1-Distill-Qwen-7B | 76.50 | 81.25 | 68.03 | 78.72 | 78.80 |
| Qwen2.5-7B-Instruct (base) | **79.30** | 85.35 | 88.22 | 88.67 | 85.50 |
| CoT-SFT-7B | 73.20 | 83.25 | 64.70 | 68.12 | 73.17 |
| **G1-7B** | 79.20 | **86.20** | 87.98 | **91.88** | **87.29** |

Table 8: Test accuracy (%) on reasoning benchmarks beyond graph-related tasks.

| Model | GSM8K | MATH | MMLU-pro |
|---|---|---|---|
| Llama-3.2-3B-Instruct | 71.03 | 42.40 | 13.50 |
| Qwen2.5-3B-Instruct (base) | **81.95** | **62.20** | **38.53** |
| CoT-SFT-3B | 75.36 | 56.00 | 34.85 |
| **G1-3B** | 79.30 | 61.80 | 37.11 |
| Llama-3.1-8B-Instruct | 74.45 | 44.80 | 32.02 |
| DeepSeek-R1-Distill-Qwen-7B | 86.03 | **87.20** | 37.21 |
| Qwen2.5-7B-Instruct (base) | 86.27 | 69.80 | 45.75 |
| CoT-SFT-7B | 83.85 | 65.80 | 44.79 |
| **G1-7B** | **87.49** | 71.80 | **48.56** |

and structural information of neighboring nodes. These benchmarks emphasize the model's ability to leverage both local graph structure and textual attributes for effective prediction.

**Results.** As shown in Table 7, our model G1 significantly outperforms both open-source and distilled baselines across tasks and model sizes. In the 3B model category, G1-3B surpasses the base model (Qwen2.5-3B-Instruct) by a large margin—especially in link prediction on Cora (+16.82%) and node classification on PubMed (+8.8%). In the 7B model category, G1-7B achieves the highest average score of 87.29%, ranking first on PubMed dataset in both node classification and link prediction tasks. Overall, G1 consistently demonstrates strong generalization across real-world graph tasks where graph-text reasoning is required.

### 5.2.3 G1's Reasoning Ability beyond Graphs

**Setup.** We next extend our investigations of G1's abilities beyond graph-based tasks. We consider two mathematics benchmarks, GSM8K [7] and MATH [16]. Additionally, we include MMLU-Pro [43], which is a massive multi-task benchmark covering disciplines such as chemistry, economics, and computer science. We believe the three benchmarks collectively provide a comprehensive assessment of G1's reasoning capabilities.

**Results.** In table 8, we first notice that the CoT-SFT training on graph reasoning trajectories leads to a non-negligible degradation in general abilities, which could be attributed to the fact that SFT *memorizes* pattern instead of incentivizing truly generalizable skills [4]. Remarkably, the subsequent reinforcement learning stage—despite being trained exclusively on graph tasks—restores the reasoning abilities of both the 3B and the 7B model. G1-7B even surpasses the performance of the initial Qwen-7B checkpoint in all of the three benchmarks (87.49% v.s. 86.27% for GSM8K, 72.8% v.s. 69.8% for MATH, and 48.56% v.s. 45.75% for MMLU-pro). Interestingly, G1-7B also outperforms Qwen-7B-Instruct on several non-STEM tasks like Economy (68.76 v.s. 46.87), which are intuitively less related to graph reasoning (see Appendix I.2 for full MMLU-Pro results). We further provide a detailed analysis on the transferability of G1 to mathematics tasks in Appendix H, showing G1 mainly improves the numerical calculation and utilization of known information.

## 5.3 Training Analysis

In this section, we further analyze the influence of two training factors on `G1`'s reasoning performance.

**Data Mixture.** In Table 2, we observe that although `G1-3B` achieves strong overall performance, it is outperformed by Direct-SFT-3B on the *Hard* and *Challenging* subsets. We hypothesize that this gap arises from imbalanced reward signals across different difficulty levels during RL training. Since correct rollouts are much easier to obtain on simpler tasks, the policy tends to allocate more of its constrained probability ratios as well as KL budget to optimize for *Easy* and *Medium* tasks, thereby maximizing the overall rewrad. To test this hypothesis, we introduce `G1-Hard-3B`, which is trained exclusively on *Hard* and *Challenging* tasks during RL. As shown in Table 9, this model achieves the highest accuracy on *Hard* (48.50%) and *Challenging* (17.43%) tasks, surpassing both `G1` and Direct-SFT. These results support our claim, suggesting that the suboptimal performance of `G1-3B` on challenging tasks is a natural consequence of the uniformly weighted reward function, rather than a shortcoming of `G1` training pipeline. Notably, despite being trained only on hard tasks, `G1-Hard-3B` also generalizes to *Easy* and *Medium* tasks (69.36% and 70.64%), far exceeding the baseline Qwen2.5-3B-Instruct. This indicates that learning to solve difficult tasks confers transferable reasoning skills that benefit performance on simpler problems. To better balance the optimization process across difficulty levels, we further explore reward-weighting strategies in Appendix J.

Table 9: Test accuracy (%) on our benchmark. ⋆ denotes the tasks are excluded in model training. `G1-Hard-3B` is only RL-trained on Hard and Challenging tasks.

| Category | Model | Easy | Medium | Hard | Challenging | Average |
|---|---|---|---|---|---|---|
| **Base Model** | Qwen2.5-3B-Instruct | 45.71 | 30.18 | 9.44 | 1.29 | 22.72 |
| | Direct-SFT-3B | 74.43 | 75.27 | 43.69 | 14.43 | 53.78 |
| **Ours** | `G1-3B` | **94.86** | **84.64** | 41.25 | 7.57 | **59.76** |
| | `G1-**Hard-3B**` | 69.36⋆ | 70.64⋆ | **48.50** | **17.43** | 53.30 |

**SFT Warmup.** We study the role of SFT as a cold-start mechanism for RL, evaluating its impact on both performance and response behavior. To isolate the effect of SFT, we compare two variants: `G1-Zero-3B` that is directly trained from the base model Qwen2.5-3B-Instruct with RL, and `G1-3B` that initializes RL from the CoT-SFT checkpoint. As shown in Figure 1, training RL directly from the base model achieves surprisingly strong performance, aligning with recent findings in Deepseek-R1-Zero [13]. Meanwhile, initializing RL with CoT-SFT provides clear and consistent improvements across all difficulty levels, with an average accuracy of 59.8% compared to 50.1% of `G1-Zero-3B`. Besides, we notice that relative improve-

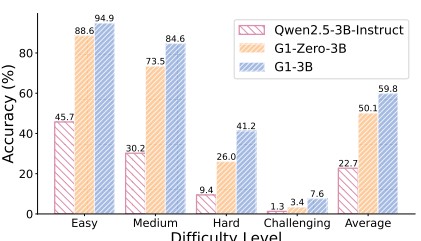

Figure 1: Test accuracy comparison of `G1`-3B and `G1-Zero-3B` on our benchmark. We include results for -7B in Appendix C.

ments become larger as the difficulty increases. In addition to performance gains, we also observe that models initialized by CoT-SFT present more precise reasoning patterns, illustrated by the case study in the following section.

## 5.4 Understanding the Benefits of RL Training for Graph Reasoning

To understand how RL training helps graph reasoning, we take *shortest path* (a Hard task) as a case study. Specifically, we study the behaviors of three models: Qwen2.5-3B-Instruct (base), `G1-Zero-3B` (RL only), and `G1-3B` (SFT & RL).

We identify three primary approaches adopted by the models to solve the problem: 1) Breadth-First Search (BFS), 2) Dijkstra's algorithm, and 3) Intuitive deductions. Figure 3a shows the distribution of these approaches alongside their corresponding accuracies for Qwen2.5-3B-Instruct. On *unweighted* graphs, BFS is the most efficient method and yields the highest performance. In contrast, Dijkstra's algorithm is best suited for *weighted* graphs, where it correctly accounts for edge costs. However, its reliance on a min-priority queue and a distance list introduces computational complexity, which appears to challenge Qwen2.5-3B-Instruct and results in its lowest observed accuracy. For example, as shown in Figure 2 (left), the model falsely states that node 4 has no edges (node 4 is connected

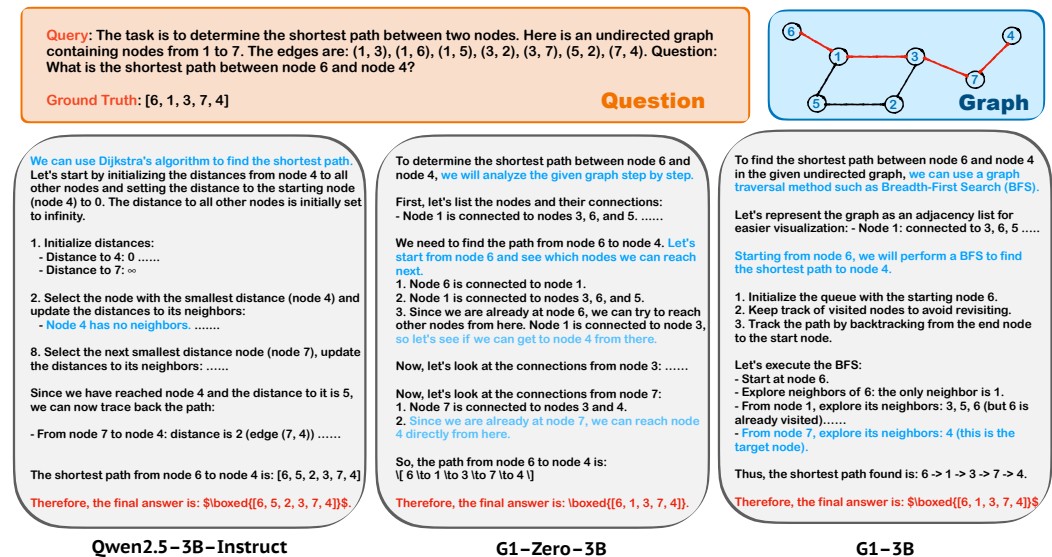

Figure 2: An intuitive illustration of the differences in solution strategies employed by Qwen2.5-3B-Instruct, `G1-Zero-3B`, and `G1-3B` for a shortest path problem.

to node 7) while updating the distance list. Interestingly, intuitive approaches—where the model attempts to visually estimate or heuristically trace paths—can also produce correct answers by a noticeable accuracy, particularly on small graphs.

We proceed by observing that RL training significantly reshapes the models' graph reasoning strategies: RL-trained models largely abandon Dijkstra and prefer a combination of BFS and intuitive search. As shown in Figure 3b and Figure 2 (middle), `G1-Zero-3B` navigates the graph in a manner akin to human heuristics—sequentially checking neighbors and adjusting paths dynamically. `G1-3B` primarily adopts a neat BFS-style algorithm as in Figure 3b and Figure 2 (right), executing it with high precision, occasionally resorting to intuitive strategies for simple graphs. To conclude, our case study highlights how RL training enhances graph reasoning by guiding LLMs toward more model-aware strategies that are adaptive to their inherent capabilities [47].

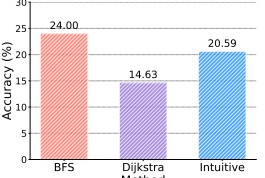

(a) The accuracy of different graph reasoning patterns for shortest path on Qwen2.5-3B-Instruct.

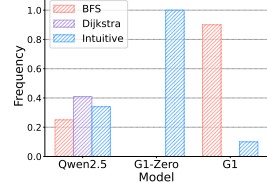

(b) Frequency of different graph reasoning patterns for Qwen2.5-3B-Instruct, `G1-Zero-3B` and `G1-3B`.

Figure 3: Reasoning patterns for the shortest path task.

# 6 Discussion

In this paper, we explored the use of Reinforcement Learning to improve LLMs' reasoning abilities on graph reasoning and demonstrate significant improvements across a spectrum of tasks with various difficulty levels, showing that graph reasoning of LLMs can be elicited via RL training (even with only 300 steps). We also comprehensively evaluate the transferability of RL-trained models to unseen graph reasoning tasks, real-world graph tasks, and general reasoning tasks, observing strong zero-shot generalization. These results support our hypothesis that training LLMs on diverse synthetic graph-theoretic tasks via RL offers a scalable, generalizable path toward robust graph reasoning. As a first step, this approach may guide the development of efficient, general-purpose graph reasoners.

In future work, we aim to explore dynamic difficulty scheduling during RL training to address the sample inefficiency issue. On a broader scale, we also plan to extend our approach to the following scenarios: (1) Handling larger graphs with thousands of nodes, aligning with the long-context reasoning challenges. (2) Incorporating visual inputs (*e.g.*, images depicting graphs) to enhance real-world applicability. (3) Adapting `G1` to more practical domains such as logistics, knowledge graph reasoning, and tabular problem-solving, where structured reasoning is critical.

## 7 Limitations

While G1 demonstrates significant improvements in graph reasoning through RL training, it inherits GRPO's sample inefficiency and requires extensive rollouts for challenging tasks (*e.g.*, NP-hard problems), which might be solved with dynamic difficulty scheduling during training. Although G1 demonstrates strong generalization to real-world graph tasks, *e.g.*, node classification and link prediction, its generalization to highly domain-specific applications (*e.g.*, molecular property prediction) and other structured data (*e.g.*, tabular, time-series) remains untested.

## Social Impacts

G1's graph reasoning capabilities offer practical benefits for education (e.g., adaptive graph theory tutoring) and scientific research. While the technology could amplify existing biases in training data, these risks can be mitigated through careful dataset design and human oversight. When developed responsibly, G1 has the potential to support human experts in solving complex graph problems.

## Acknowledgement

Yisen Wang was supported by Beijing Natural Science Foundation (L257007), Beijing Major Science and Technology Project under Contract no. Z251100008425006, National Natural Science Foundation of China (92370129, 62376010), Beijing Nova Program (20230484344, 20240484642), and State Key Laboratory of General Artificial Intelligence. Yifei Wang and Stefanie Jegelka were supported in part by the NSF AI Institute TILOS (NSF CCF-2112665), and an Alexander von Humboldt Professorship.

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

# A Training Details

## A.1 Rejection Sampling

We randomly extract a subset with 100 examples per task from the training dataset, and use Qwen2.5-32B-Instruct to sample on the subset for k = 8 times with a temperature of 1.0. We filter the responses by keeping the reasoning steps that lead to the right answer. If the task is difficult and the filtered responses are insufficient, we resample the subset with a different random seed and repeat the process above. In the end, we obtain around 4,500 training examples ($\sim$90 per task) for the SFT phase.

## A.2 Supervised Fine-tuning

The detailed training configurations of Naive SFT and RFT are presented in Table 10.

Table 10: Training configurations of Naive-SFT and RFT. In this table, batch size is abbreviated to BSZ, Max-Length refers to the maximum response length during training and Data Num. reports the number of training examples.

| Setting | LR | Weight Decay | BSZ | Max-Length | Data Num. | Epoch |
|---|---|---|---|---|---|---|
| Naive-SFT | 1e-5 w/ 1% warm-up | 1e-2 | 64 | 512 | 98.7k | 1 |
| RFT | 1e-5 w/ 1% warm-up | 1e-2 | 64 | 3072 | 4.4k | 2 |

## A.3 Reinforcement Learning

**Configurations for training and evaluation**. Our experiments primarily adopt Qwen-2.5-3B/7B-Instruct [32] for their moderate sizes and strong reasoning performance. For GRPO training, we set $\epsilon$ to be 0.02, $\beta$ to be 0.001, group size $G$ to be 5, and context length to be 4096 unless otherwise specified. We additionally incorporate an entropy loss of weight 0.001 to encourage the policy to explore. Lastly, we train the models on 8xA800 GPUs with batch size of 512. During evaluation, we use the vLLM [22] engine for efficient inference. For DeepSeek-R1-Distill-Qwen-7B, we set the maximum token generation length to 4096 tokens except for DeepSeek-R1-Distill-Qwen-7B, which is extended to 30768 for its prolonged thinking process. Sampling is configured with a temperature of 0.6, top-p of 0.95, and top-k of 30.

The detailed RL training configurations are presented in Table 11.

Table 11: Training configurations for Naive-SFT and RFT. For abbreviation, we refer the coefficient for entropy loss as Ent. in this table. We report (batch size)/(number of gradient accumulation steps) in the BSZ column, and the temperature for on-policy sampling as $T$.

| Model | LR | $\epsilon$ | $|G|$ | $\beta$ | $\gamma$ | $T$ | Ent. | BSZ | Max-Length | Data Num. | Steps |
|---|---|---|---|---|---|---|---|---|---|---|---|
| RL-3B | 1e-6 | 0.2 | 5 | 1e-3 | 1.0 | 1.0 | 1e-3 | 512/4 | 4096 | 98.7k | 300 |
| SFT-RL-3B | 1e-6 | 0.2 | 5 | 1e-3 | 1.0 | 1.0 | 1e-3 | 512/4 | 4096 | 98.7k | 300 |
| SFT-RL-Hard-3B | 1e-6 | 0.2 | 16 | 5e-4 | 1.0 | 1.0 | 5e-4 | 512/8 | 8192 | 49.3k | 150 |
| SFT-RL-7B | 1e-6 | 0.2 | 5 | 1e-3 | 1.0 | 1.0 | 1e-3 | 512/8 | 4096 | 98.7k | 300 |

# B Evaluation Details

## B.1 Benchmark Introduction

**GraphWiz** [3]. GraphWiz employs the Erdős-Rényi (ER) model to generate random graphs and describe graphs in the edge-list formation like $(u, v)$. The tasks include four linear complexity tasks, *Connectivity*, *Cycle Detection*, *Bipartite Graph Checking*, and *Topological Sort*; three polynomial complexity tasks, *Shortest Path*, *Maximum Triangle Sum*, and *Maximum Flow*; and two NP-Complete tasks: *Hamilton Path* and *Subgraph Matching*. A prompt example is shown in the following:

**Maximum Triangle Sum Example in GraphWiz**

Find the maximum sum of the weights of three interconnected nodes. In an undirected graph, [i, k] means that node i has the weight k. (i,j) means that node i and node j are connected with an undirected edge. Given a graph, you need to output the maximum sum of the weights of three interconnected nodes. Q: The nodes are numbered from 0 to 4, weights of nodes are: [0, 8] [1, 5] [2, 3] [3, 6] [4, 3], and the edges are: (0, 4) (0, 3) (0, 1) (1, 3) (1, 2) (3, 4). What is the maximum sum of the weights of three nodes?

**Node Classification and Link Prediction** [44]. We adopt the benchmarks introduced by Wang et al. [44], which are constructed by subsampling from the widely used Cora and PubMed citation graphs. Each instance includes a description of the target node (or node pair) containing the paper ID and title, along with the textual and structural information of neighboring nodes. For node classification, we consider two cases that the description includes the attributes of the target node and those of its 2-hop neighbors, with or without labels. For link prediction, we consider two cases where target nodes are described using their own node attributes along with those of their 2-hop neighbors (excluding the other targeting node), with or without titles. For each task, we randomly sample 2,000 examples per case from the benchmark and report the average performance. A representative example for node classification is shown below:

**Node Classification Example**

You are a good graph reasoner. Give you a graph language that describes a graph structure and node information from pubmed dataset. You need to understand the graph and the task definition and answer the question.

## Target node: Paper id: 10695 Title: Haplotype structures and large-scale association testing of the 5' AMP-activated protein kinase genes PRKAA2, PRKAB1, and PRKAB2 [corrected] with type 2 diabetes.

Known neighbor papers at hop 1 (partial, may be incomplete):
Paper id: 1155 Title: Computational disease gene identification: a concert of methods prioritizes type 2 diabetes and obesity candidate genes. Label: Type 2 diabetes

Known neighbor papers at hop 2 (partial, may be incomplete):
Paper id: 9816 Title: Mitochondrial dysfunction and type 2 diabetes. Label: Type 2 diabetes
Paper id: 1683 Title: A genome-wide search for type II diabetes susceptibility genes in Chinese Hans. Label: Type 2 diabetes
Paper id: 9916 Title: Genomewide search for type 2 diabetes-susceptibility genes in French whites: evidence for a novel susceptibility locus for early-onset diabetes on chromosome 3q27-qter and independent replication of a type 2-diabetes locus on chromosome 1q21-q24.
Paper id: 3793 Title: Association of amino acid variants in the activating transcription factor 6 gene (ATF6) on 1q21-q23 with type 2 diabetes in Pima Indians. Label: Type 2 diabetes
Paper id: 4788 Title: Altered glycolytic and oxidative capacities of skeletal muscle contribute to insulin resistance in NIDDM. Label: Type 2 diabetes

Please predict the most likely type of the Target node. Your answer should be chosen from: Type 1 diabetes Type 2 diabetes Experimentally induced diabetes

**GraphArena** [38]. GraphArena samples subgraphs from real-world graphs, including knowledge graphs, social networks, and molecular structures. The tasks include four polynomial-time tasks, *Common Neighbor*, *Shortest Distance*, *Connected Component*, *Graph Diameter*, and six NP-complete tasks, *Maximum Clique Problem (MCP)*, *Maximum Independent Set (MIS)*, *Minimum Vertex Cover (MVC)*, *Maximum Common Subgraph (MCS)*, *Graph Edit Distance (GED)*, and *Traveling Salesman Problem (TSP)*. Each problem is contextualized within the real-world setting of the graph with an example presented as below:

> **Connected Component Example in GraphArena**
>
> You are required to identify all connected components in the given social network and output one representative node from each component. Within a connected component, any node can be reached from any other node through the edges in the graph. Different connected components are isolated from each other.
>
> **Problem to Solve**
> - Names in the network: Veronica Garcia, Katherine Brennan, Angel Chavez, Steven Martin, Brett Johnson, Megan Banks, Julia Dominguez, Rachel Mitchell - Fiendship connections: Veronica Garcia to Brett Johnson, Veronica Garcia to Megan Banks, Katherine Brennan to Brett Johnson, Katherine Brennan to Megan Banks, Angel Chavez to Megan Banks, Angel Chavez to Rachel Mitchell, Steven Martin to Megan Banks, Brett Johnson to Megan Banks, Megan Banks to Julia Dominguez, Megan Banks to Rachel Mitchell.
>
> Identify all connected components in this network. Note that for each connected component, you should only output one of its nodes. Present your answer in the following format: [UserA, UserB, UserC, UserD, ...]

**GSM8K** [6]. GSM8K is a dataset of 8.5K high quality linguistically diverse grade school math word problems created by human problem writers. We report the accuracies on the 1K test problems and the dataset is downloaded via `https://huggingface.co/datasets/openai/gsm8k`.

> **Example in GSM8K**
>
> Natalia sold clips to 48 of her friends in April, and then she sold half as many clips in May. How many clips did Natalia sell altogether in April and May?

**MATH500.** The dataset contains a subset of 500 problems from the MATH benchmark that OpenAI created in their Let's Verify Step by Step paper [25]. We download the dataset via `https://huggingface.co/datasets/HuggingFaceH4/MATH-500`.

> **Example in MATH500**
>
> Let $z = 2 + \sqrt{2} - (3 + 3\sqrt{2})i$, and let $c = 2 - 3i$. Let $w$ be the result when $z$ is rotated around $c$ by $\frac{\pi}{4}$ counter-clockwise.
> [asy]
> unitsize(0.6 cm);
> pair C, W, Z;
> Z = (2 + sqrt(2), -3 - 3*sqrt(2));
> C = (2,-3);
> W = rotate(45,C)*(Z);
> draw(Z--C--W);
> dot("$c$", C, N);
> dot("$w$", W, SE);
> dot("$z$", Z, S);
> label("$\frac{\pi}{4}$", C + (0.6,-1));
> [/asy]
> Find $w$.

**MMLU-Pro.** MMLU-Pro is enhanced version of the Massive Multitask Language Understanding benchmark. It covers a wide range of disciplines, including Math, Law, Engineering, Health, Phycology, etc. We download the dataset via `https://huggingface.co/datasets/TIGER-Lab/MMLU-Pro/viewer/default/test?q=Health&row=5903`.

> **Health Example in MMLU-pro**
>
> Question: Food supplements, including trace minerals and vitamins are frequently advertised with promising health benefits. Which of the following substance could be consumed in excess, i.e. well above the recommended daily requirement?
>
> Options: [ "Vitamin C", "Vitamin D", "Zinc", "Vitamin A" ]

## B.2 Inference Configuration

For inference, we adopt the vLLM framework [22]. We set the temperature to be 0.06 and the context window to be 4096 for our evaluations unless otherwise specified.

## B.3 Prompt and Answer Extraction

To facilitate answer extraction, we adopt the prompt shown in B.3 to guide the models to reason step by step and place their answers within \boxed{}. We extract the last \boxed{} shown in the model responses and do necessary format normalizations to retrieve the answer, which includes operations like converting LaTeX-style fraction numbers to float numbers.

> **Problem Instructions**
>
> {Question Description}
> Approach the problem methodically. Ensure all conclusions are based on precise calculations and logical deductions. Feel free to explore various solution methods and cross-check results for consistency. Maintain dynamic thinking and always verify each step of your reasoning.
> Present the final answer in \boxed{} format, like this: $\boxed{ANSWER}$, where ANSWER is the final result or expression.
> Think carefully and break down the problem step by step.

## C Comparison between `G1`-Zero-7B and `G1`-7B

In Section 5.3, we study the role of SFT as a cold-start mechanism for RL by comparing two variants: `G1`-Zero-3B that is directly trained from the base model Qwen2.5-3B-Instruct with RL, and `G1`-3B that initializes RL from the CoT-SFT checkpoint. We observe that `G1`-Zero-3B already achieves surprisingly strong performance, while `G1`-3B presents clear and consistent improvements across all difficulty levels. Here, we provide additional results for comparing `G1`-Zero-7B and `G1`-7B. As shown in Figure 4, for *Easy* and *Medium* tasks, the benefit brought by CoT-SFT initialization is marginal, with `G1`-Zero-7B (96.9%) even surpassing `G1`-7B (95.1%) on *Easy* tasks. However,

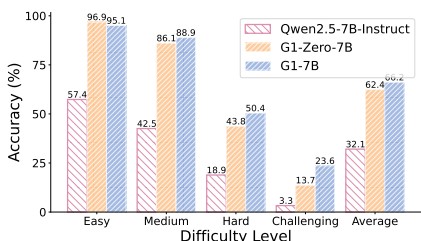

Figure 4: Test accuracy comparison of G1-7B and G1-Zero-7B on our benchmark.

on *Hard* and *Challenging* tasks, CoT-SFT as a preliminary step has definite benefits by improving `G1`-Zero-7B from 13.7% to 23.6% on *Challenging* tasks. This observation agrees with the case in -3B. Moreover, the average gap between `G1`-Zero-7B and `G1`-7B is less than -3B case, indicating `G1`-7B can possibly be further improved with CoT-SFT generated by a stronger teacher model rather than Qwen2.5-32B-Instruct. We leave this exploration for further work.

## D Transferability of `G1` to Larger Graphs

To verify the transferability of `G1` to larger graphs, we construct a new test set of 5,000 graphs with 36-100 nodes, with other settings kept the same, which ensures there is no overlap between training and test data. Table 12 shows that both `G1`-3B and `G1`-7B achieve strong zero-shot generalization to these larger graphs without additional training, significantly outperforming the baselines across

difficulty levels. These results demonstrate our method's effective scalability beyond the original training distribution.

Currently, we limit our analysis to smaller graphs because of the context window limit of underlying LLMs (*e.g.*, Qwen2.5-7B-Instruct). The token count scales quadratically with the number of nodes. As shown in Table 13, a 200-node graph often exceeds 32k tokens, surpassing the maximum effective context window of many open-source LLMs [17]. Long-context understanding is actively studied in the LLM literature, and there are some cutting-edge proprietary variants (*e.g.*, OpenAI's GPT-4.1) supporting inputs of over 1M tokens (though graphs of 2000 nodes need ~4M). Due to computational constraints (demanding a huge GPU memory), it is hard for us to evaluate on a very long context. We believe our approach can be scaled to larger graphs with the rapid progress of long-context studies.

Table 12: Zero-shot generalization (accuracy in percentage) of `G1` to larger graphs with 36-100 nodes.

|  | Easy | Medium | Hard | Challenging | Average |
|---|---|---|---|---|---|
| Qwen2.5-3B-Instruct | 27.98 | 28.53 | 5.26 | 0.29 | 16.74 |
| **G1-3B** | **79.39** | **65.66** | **18.46** | **3.74** | **44.29** |
| Qwen2.5-7B-Instruct | 37.86 | 41.56 | 9.17 | 1.17 | 23.94 |
| **G1-7B** | **76.65** | **70.67** | **23.16** | **5.22** | **46.46** |

Table 13: Token numbers of graph with different node sizes. We generate random graphs by Erdős–Rényi model with an edge probability of 0.2. For each node number, we generate 10 graphs and report the mean of the token numbers. For tokenization, we utilize the tokenizer of Qwen2.5-Instruct.

| **Node Number** | 30 | 50 | 100 | 200 | 500 | 1000 | 2000 |
|---|---|---|---|---|---|---|---|
| **Token Number** | 641.4 | 1941.5 | 7842.8 | ~35k | ~230k | ~1M | ~4M |

# E   GPT-4o Results on `Erdős`

Due to the cost budget, we randomly sample 20 examples per task from `Erdős`'s evaluation set to construct a subset of 1,000 samples. As shown in Table 14, `G1-3B` performs comparably to GPT-4o on average (57.37% vs. 55.13%), while `G1-7B` surpasses GPT-4o across all difficulty levels, exceeding it by over 10% on average. This comparison further validates the strong graph reasoning capabilities of the `G1` models.

Table 14: Test accuracy%) of GPT-4o and `G1` variants on a subset of Erdős.

|  | Easy | Medium | Hard | Challenging | Average |
|---|---|---|---|---|---|
| GPT-4o-2024-11-20 | 82.50 | 81.82 | 44.06 | 12.14 | 55.13 |
| **G1-3B** | **96.43** | 85.45 | 41.88 | 5.71 | 57.37 |
| **G1-7B** | **96.43** | **88.64** | **52.50** | **23.57** | **65.29** |

# F   Results of `G1`-**Zero-32B**

To demonstrate the scalability of our approach, we extended our training methodology to develop G1-Zero-32B from Qwen2.5-32B-Instruct. The results showcase substantial improvements across all difficulty levels while maintaining computational efficiency. As shown in Table 15, G1-Zero-32B demonstrates remarkable improvements on the Erdősbenchmark across all difficulty categories. The model achieves a **27.96%** improvement in average accuracy (from 47.10% to 75.06%), with particularly notable gains in harder categories: +31.87 points on Hard problems and +26.43 points on Challenging problems. These results indicate that our method scales effectively to larger models while maintaining consistent performance improvements across varying problem complexities.

Furthermore, Table 16 demonstrates that `G1-Zero-32B` not only preserves but slightly enhances mathematical performance on standard benchmarks. The model shows modest improvements on both

GSM8K (+0.08 points) and MATH (+4.00 points), confirming that our reasoning-focused training does not compromise existing mathematical capabilities and may even provide synergistic benefits. The training process is completed in 40 hours on 32×A100 GPUs, demonstrating the practical feasibility of scaling our approach to larger model architectures without prohibitive computational costs.

Table 15: Test accuracy (%) of G1-Zero-32B and Qwen2.5-32B-Instruct on Erdős.

|  | Easy | Medium | Hard | Challenging | Average |
|---|---|---|---|---|---|
| Qwen2.5-32B-Instruct | 70.57 | 68.73 | 33.38 | 9.00 | 47.10 |
| G1-Zero-32B | **97.79** | **93.00** | **65.25** | **35.43** | **75.06** |

Table 16: Test accuracy (%) of G1-Zero-32B and Qwen2.5-32B-Instruct on math tasks.

|  | GSM8K | MATH |
|---|---|---|
| Qwen2.5-32B-Instruct | 90.67 | 76.80 |
| G1-Zero-32B | **90.75** | **80.80** |

# G   Experiments for Robustness Verification

## G.1   Multi-runs Robustness

To rigorously evaluate robustness, we conducted 32 repeated runs with different random seeds. The results in Table 17 demonstrate consistently small standard deviations (<1% across all models and difficulty levels), confirming the stability of our method against potential randomness in LLM outputs.

Table 17: Test accuracy (%) for 32 runs with different random seeds.

|  | Easy | Medium | Hard | Challenging |
|---|---|---|---|---|
| Qwen2.5-3B-Instruct | $45.65 \pm 0.51$ | $30.88 \pm 0.36$ | $10.36 \pm 0.21$ | $1.54 \pm 0.29$ |
| G1-3B | $94.96 \pm 0.32$ | $83.22 \pm 0.24$ | $41.48 \pm 0.40$ | $7.96 \pm 0.64$ |
| Qwen2.5-7B-Instruct | $57.50 \pm 0.13$ | $44.92 \pm 0.03$ | $19.90 \pm 0.31$ | $3.45 \pm 0.22$ |
| G1-7B | $95.66 \pm 0.12$ | $88.89 \pm 0.16$ | $50.76 \pm 0.53$ | $24.46 \pm 0.84$ |

## G.2   Prompt Robustness

For prompt robustness, we rigorously test prompt sensitivity by having GPT-4o generate three semantically equivalent prompt variants. Table 18 shows minimal performance variance (<1.5% standard deviation) across all models and difficulty levels, confirming our benchmark's stability to phrasing changes.

# H   Analysis of `G1`'s Transferbility to Mathematics Tasks

We compare the generation results of `G1-7B` and Qwen2.5-7B-Instruct on the math benchmarks case by case. As a summary, `G1-7B` improves over Qwen2.5-7B-Instruct on more accurate *numerical calculation* (Instance 1) and more comprehensive *utilization of the given information* (Instance 2). It's reasonable since the training of `G1` includes complex numerical calculations (e.g., shortest path) and careful processing of questions (e.g., traversing the whole graph descriptions to find key edges).

In fact, the transferability between different reasoning domains has also been observed in other works and is actively under research. For example, Chu et al. [5] demonstrates that reinforcement learning exhibits strong generalization even under distribution shifts, as it encourages models to learn robust, task-agnostic reasoning strategies rather than overfitting to surface-level patterns. Additionally, RL training has been shown to foster mutual reinforcement across seemingly distinct tasks, *e.g.*, optimizing on puzzles can help reasoning on math [24]. Similarly, due to the similarity between math and graph problems, they would also require similar reasoning skills (such as guessing, backtracking,

Table 18: Test accuracy (%) on different prompts, mean and standard deviation reported.

|  | Easy | Medium | Hard | Challenging |
|---|---|---|---|---|
| Qwen2.5-3B-Instruct | $44.26 \pm 0.63$ | $30.52 \pm 1.08$ | $10.71 \pm 0.41$ | $1.24 \pm 0.07$ |
| G1-3B | $94.86 \pm 0.69$ | $83.03 \pm 0.42$ | $41.08 \pm 1.35$ | $9.67 \pm 1.05$ |
| Qwen2.5-7B-Instruct | $56.48 \pm 1.17$ | $56.48 \pm 1.17$ | $18.81 \pm 0.79$ | $3.10 \pm 0.18$ |
| G1-7B | $95.48 \pm 0.18$ | $88.39 \pm 0.67$ | $51.27 \pm 0.13$ | $25.14 \pm 1.04$ |

and verification). Therefore, solving graph problems could have positive transferability to math domains.

---

**Instance 1 (Calculation Correctness)**

**Question**: Toula went to the bakery and bought various types of pastries. She bought 3 dozen donuts, which cost 68 dollars per dozen, 2 dozen mini cupcakes, which cost 80 dollars per dozen, and 6 dozen mini cheesecakes for 55 dollars per dozen. How much was the total cost?

**Ground Truth**: 694

**Answer of Qwen2.5-7B-Instruct**: - Calculate the cost of the donuts: Toula bought 3 dozen donuts. The cost per dozen donuts is 68 dollars. Therefore, the total cost for the donuts is: $3 \text{ dozen} \times 68 \text{ dollars/dozen} = 180 \text{ dollars}$.
- Calculate the total cost: Add the costs of the donuts, mini cupcakes, and mini cheesecakes: $180 \text{ dollars} + 160 \text{ dollars} + 330 \text{ dollars} = 670 \text{ dollars}$.

**Answer of G1-7B**: - Cost of Donuts: Toula bought 3 dozen donuts. The cost per dozen donuts is 68 dollars. - Therefore, the total cost for the donuts is: $3 \text{ dozen} \times 68 \text{ per dozen} = 3 \times 68 = 204$.
- Now, we add up the costs of all the pastries to find the total cost: \$204 (donuts) + \$160 (mini cupcakes) + \$330 (mini cheesecakes) = \$694.

---

**Instance 2 (Utilization of Known Conditions)**

**Question**: On Mondays, Wednesdays, and Fridays, college student Kimo has three 1-hour classes each day. On Tuesdays and Thursdays, he has two 2-hour classes each day. In one semester, there are 16 weeks of school. In one semester, how many hours does Kimo spend attending classes?

**Ground Truth**: 272

**Answer of Qwen2.5-7B-Instruct**: - Calculate the number of hours Kimo spends in classes each week: On Mondays, Wednesdays, and Fridays, Kimo has 3 classes, each lasting 1 hour. $3 \text{ classes} \times 1 \text{ hour/class} = 3 \text{ hours}$. On Tuesdays and Thursdays, Kimo has 2 classes, each lasting 2 hours. $2 \text{ classes} \times 2 \text{ hours/class} = 4 \text{ hours}$. Therefore, the total number of hours Kimo spends in classes each week is: $3 \text{ hours} + 4 \text{ hours} = 7 \text{ hours}$. - Calculate the number of hours Kimo spends in classes in one semester: There are 16 weeks in the semester. The total number of hours Kimo spends in classes in one semester is: $7 \text{ hours/week} \times 16 \text{ weeks} = 112 \text{ hours}$.

**Answer of G1-7B**: - Step 1: Calculate Weekly Class Hours. - Mondays, Wednesdays, and Fridays: Each day, Kimo has 3 classes, each lasting 1 hour. Therefore, for each of these days, the total class time is: $3 \text{ classes/day} \times 1 \text{ hour/class} = 3 \text{ hours/day}$. Since there are 3 days (Monday, Wednesday, Friday): $3 \text{ hours/day} \times 3 \text{ days} = 9 \text{ hours}$. - Tuesdays and Thursdays: Each day, Kimo has 2 classes, each lasting 2 hours. Therefore, for each of these days, the total class time is: $2 \text{ classes/day} \times 2 \text{ hours/class} = 4 \text{ hours/day}$. Since there are 2 days (Tuesday, Thursday) $4 \text{ hours/day} \times 2 \text{ days} = 8 \text{ hours}$. - Total Weekly Class Hours: Adding the hours from all the days: $9 \text{ hours} + 8 \text{ hours} = 17 \text{ hours/week}$. - Step 2: Calculate Total Class Hours in One Semester. There are 16 weeks in the semester. Therefore, the total class hours in one semester is: $17 \text{ hours/week} \times 16 \text{ weeks} = 272 \text{ hours}$.

# I Detailed Experimental Results

## I.1 Detailed Results for GraphWiz

We present the test accuracy for each task in the GraphWiz benchmark in Table 19. `G1-7B` achieves the highest overall accuracy (57.11%) among all models and reaches the top in 5/7 tasks. It outperforms DeepSeek-R1-Distill-Qwen-7B (51.86%) and even models specifically trained on GraphWiz data such as GraphWiz-7B-RFT (49.61%). Moreover, the smaller variant `G1-3B` ranks first on all tasks among models of similar parameters, surpassing the base model (Qwen2.5-3B-Instruct) by 13.64% on average and achieves comparable performance with DeepSeek-R1-Distill-Qwen-7B. The results in the GraphWiz benchmark verify the strong zero-shot generalization ability of our `G1` models.

Table 19: Test accuracy (%) on the GraphWiz benchmark.

| Model | cycle | connect | bipartite | topology | shortest | triangle | flow | hamilton | subgraph | Avg. |
|---|---|---|---|---|---|---|---|---|---|---|
| Llama-3.2-3B-Instruct | 32.00 | 53.75 | 25.75 | 7.50 | 2.75 | 3.75 | 2.50 | 38.25 | 12.00 | 19.80 |
| Qwen2.5-3B-Instruct (base) | 58.00 | 60.50 | 38.50 | 4.00 | 5.75 | 15.50 | 7.50 | 75.00 | **63.25** | 36.44 |
| **G1-3B** (Ours) | **91.00** | **64.00** | **64.25** | **13.00** | **14.00** | **23.25** | **43.00** | **96.00** | 42.25 | **50.08** |
| GraphWiz-RFT-7B | 88.00 | **90.25** | 72.25 | 19.75 | **28.00** | 36.75 | 24.75 | 2.50 | 84.25 | 49.61 |
| GraphWiz-DPO-7B | 86.50 | 82.25 | 71.75 | 15.00 | 26.75 | **37.00** | 45.00 | 0.00 | 79.00 | 49.25 |
| Llama-3.1-8B-Instruct | 64.75 | 81.00 | 58.75 | 11.50 | 3.50 | 4.25 | 9.25 | 19.25 | 45.00 | 33.03 |
| DeepSeek-R1-Distill-Qwen-7B | 87.00 | 90.00 | 42.75 | 11.00 | 18.25 | 36.00 | 40.00 | 84.75 | 57.00 | 51.86 |
| Qwen2.5-7B-Instruct (base) | 79.00 | 72.25 | 40.75 | 4.25 | 13.50 | 28.75 | 11.50 | 91.25 | 61.00 | 44.69 |
| **G1-7B** (Ours) | **92.00** | 80.00 | **75.75** | **24.25** | 21.00 | 29.50 | **46.25** | 95.25 | 50.00 | **57.11** |

## I.2 Detailed Results for MMLU-Pro

We present the detailed results for our evaluations on MMLU-Pro in Table 20. We first notice that although `G1` models share close accuracies with their base model on average, they excel at notably different disciplines: `G1-3B` does the best in Physics (56.18%) while `G1-7B` is good at CS (53.32%). Interestingly, RL training on graph problems in some cases improves `G1` over Qwen on non-STEM subjects such as Health (53.0% v.s. 37.65%) for 3B models and Business (62.76% v.s. 53.91%) for 7B models.

Table 20: Test accuracy (%) on the MMLU-Pro benchmark.

| Model | Physics | Chem. | Econ. | Other | Math | Philo. | History | Busi. | Psycho. | Law | Engin. | Health | CS | Bio. | Avg. |
|---|---|---|---|---|---|---|---|---|---|---|---|---|---|---|---|
| Llama-3.2-3B-Instruct | 7.18 | 14.79 | 15.91 | 13.39 | 6.50 | 13.69 | 18.54 | 11.28 | 23.91 | 15.40 | 9.89 | 14.03 | 13.25 | 9.71 | 13.51 |
| Qwen2.5-3B-Instruct (base) | 38.49 | 31.18 | **46.21** | 37.34 | **58.92** | 31.06 | 31.23 | **45.25** | **46.24** | 18.07 | 19.40 | 37.65 | **41.22** | **54.25** | **38.54** |
| CoT-SFT-3B | 35.70 | 13.99 | 32.25 | 38.72 | 53.29 | 34.41 | 25.65 | 30.04 | 18.16 | **42.71** | 28.08 | 39.22 | 36.34 | 46.03 | 34.23 |
| **G1-3B** (Ours) | **56.18** | **42.46** | 16.26 | **43.73** | 37.78 | **44.55** | **36.10** | 31.80 | 41.46 | 20.95 | **34.42** | **53.00** | 28.86 | 30.18 | 37.12 |
| Llama-3.1-8B-Instruct | 28.79 | 17.13 | 33.96 | 34.03 | 32.28 | 41.83 | 24.91 | 18.80 | 43.89 | 46.45 | 35.28 | 36.10 | 31.75 | 28.26 | 32.02 |
| DeepSeek-R1-Distill-Qwen-7B | 39.75 | 11.72 | 19.20 | 49.81 | 40.80 | 19.95 | 23.35 | 25.65 | **47.39** | 30.30 | **72.76** | 36.84 | 34.59 | 49.51 | 37.21 |
| Qwen2.5-7B-Instruct (base) | 44.17 | 48.53 | 46.87 | **55.89** | **65.80** | 21.44 | **54.53** | 53.91 | 27.04 | 49.50 | 42.64 | **53.66** | 33.27 | 35.96 | 45.75 |
| CoT-SFT-7B | 44.36 | **55.51** | 44.61 | 29.82 | 51.08 | **64.84** | 45.97 | 41.45 | 46.42 | 37.01 | 33.87 | 45.61 | 21.44 | **52.01** | 44.54 |
| **G1-7B** (Ours) | 46.43 | 51.19 | **68.76** | 40.94 | 47.70 | 53.90 | 32.40 | **62.76** | 25.61 | **49.88** | 51.50 | 51.71 | **53.32** | 36.07 | **48.56** |

## I.3 Detailed Results for GraphArena

We report the detailed results for evaluations on the easy/hard problems from GraphArena in Table 21 and Table 22 respectively. We observe that `G1` models perform equally or better compared to the other models on all tasks but *Distance*, in which `G1` performs slightly worse than the Qwen models.

## I.4 Detailed Results for Erdős

In Table 23, we show the performance for each task in Erdős for our models and baselines in detail.

Table 21: Test accuracy (%) on the **easy** problems from the GraphArena benchmark.

| Model | Connected | Diameter | Distance | Neighbor | GED | TSP | MCP | MCS | MIS | MVC |
|---|---|---|---|---|---|---|---|---|---|---|
| Llama-3.2-3B-Instruct | 8.00 | 16.00 | 15.00 | 50.00 | 9.00 | 2.00 | 15.00 | 10.00 | 7.00 | 5.00 |
| Qwen2.5-3B-Instruct (base) | 20.00 | 11.00 | **47.00** | 48.00 | **37.00** | **17.00** | 3.00 | **41.00** | 4.00 | 2.00 |
| **G1-3B** (Ours) | **52.00** | **42.00** | **47.00** | **89.00** | 30.00 | **17.00** | **27.00** | 20.00 | **32.00** | **22.00** |
| LLaMA2-7B-RFT | 0.00 | 7.00 | 1.00 | 1.00 | 4.00 | 0.00 | 0.00 | 1.00 | 0.00 | 0.00 |
| LLaMA2-7B-DPO | 0.00 | 1.00 | 0.00 | 0.00 | 3.00 | 0.00 | 0.00 | 1.00 | 0.00 | 0.00 |
| Llama-3.1-8B-Instruct | 33.00 | 29.00 | 45.00 | 81.00 | 24.00 | 14.00 | 32.00 | 18.00 | 24.00 | 20.00 |
| DeepSeek-R1-Distill-Qwen-7B | 77.00 | 41.00 | 64.00 | 82.00 | 22.00 | 30.00 | 44.00 | 40.00 | **56.00** | 17.00 |
| Qwen2.5-7B-Instruct (Ours) | 79.00 | 15.00 | **70.00** | 84.00 | 22.00 | 22.00 | 39.00 | 41.00 | 28.00 | 21.00 |
| **G1-7B** (Ours) | **86.00** | **63.00** | 62.00 | **99.00** | **30.00** | **38.00** | **52.00** | **51.00** | 50.00 | **63.00** |

Table 22: Test accuracy (%) on the **hard** problems from the GraphArena benchmark.

| Model | Connected | Diameter | Distance | Neighbor | GED | TSP | MCP | MCS | MIS | MVC |
|---|---|---|---|---|---|---|---|---|---|---|
| Llama-3.2-3B-Instruct | 0.00 | 1.00 | 7.00 | 19.00 | 3.00 | 0.00 | 0.00 | 0.00 | 0.00 | 1.00 |
| Qwen2.5-3B-Instruct (base) | 4.00 | 4.00 | **28.00** | 22.00 | **7.00** | 0.00 | **1.00** | 0.00 | 0.00 | 1.00 |
| **G1-3B** (Ours) | **19.00** | **12.00** | 25.00 | **51.00** | 3.00 | 0.00 | 0.00 | 0.00 | **1.00** | **7.00** |
| LLaMA2-7B-RFT | 0.00 | 2.00 | 0.00 | 1.00 | 0.00 | 0.00 | 0.00 | 0.00 | 0.00 | 0.00 |
| LLaMA2-7B-DPO | 0.00 | 3.00 | 0.00 | 1.00 | 1.00 | 0.00 | 0.00 | 0.00 | 0.00 | 0.00 |
| Llama-3.1-8B-Instruct | 8.00 | 4.00 | 19.00 | 54.00 | **3.00** | 1.00 | 2.00 | 0.00 | 0.00 | 7.00 |
| DeepSeek-R1-Distill-Qwen-7B | 18.00 | 4.00 | 33.00 | 36.00 | 1.00 | 0.00 | 3.00 | 0.00 | 1.00 | 4.00 |
| Qwen2.5-7B-Instruct (base) | 27.00 | 4.00 | **44.00** | 68.00 | 2.00 | 0.00 | **5.00** | 0.00 | 1.00 | 5.00 |
| **G1-7B** (Ours) | **31.00** | **27.00** | 35.00 | **84.00** | **3.00** | 0.00 | 3.00 | 0.00 | **6.00** | **39.00** |

Table 23: Accuracy comparison of models across tasks

| Task | GPT-4o | o3-mini | Llama-3B | Qwen-3B | DSFT-3B | CSFT-3B | G1-3B | Llama-8B | Qwen-7B | Math-7B | R1-7B | GWiz-R | GWiz-D | DSFT-7B | CSFT-7B | G1-7B | Llama-70B | Qwen-72B |
|---|---|---|---|---|---|---|---|---|---|---|---|---|---|---|---|---|---|---|
| node_number | 100.00 | 100.00 | 83.00 | 94.00 | 100.00 | 97.00 | 100.00 | 99.00 | 99.00 | 94.00 | 100.00 | 0.00 | 0.00 | 100.00 | 100.00 | 100.00 | 100.00 | 100.00 |
| dominating_set | 29.41 | 64.71 | 57.00 | 23.00 | 72.00 | 31.00 | 99.00 | 37.00 | 27.00 | 21.00 | 27.00 | 34.00 | 28.00 | 74.00 | 68.00 | 99.00 | 24.00 | 44.00 |
| common_neighbor | 73.68 | 73.68 | 23.00 | 44.00 | 71.00 | 71.00 | 91.00 | 56.00 | 52.00 | 48.00 | 79.00 | 0.00 | 0.00 | 76.00 | 80.00 | 97.00 | 91.52 | 89.99 |
| edge_number | 72.22 | 77.78 | 9.00 | 31.00 | 31.00 | 59.00 | 96.00 | 16.00 | 58.00 | 38.00 | 74.00 | 0.00 | 0.00 | 39.00 | 34.00 | 97.00 | 72.00 | 66.00 |
| neighbor | 84.21 | 63.16 | 26.00 | 36.00 | 87.00 | 82.00 | 91.00 | 42.00 | 65.00 | 64.00 | 94.00 | 4.00 | 2.00 | 89.00 | 89.00 | 93.00 | 93.05 | 98.53 |
| bfs | 52.17 | 17.39 | 0.00 | 3.00 | 52.00 | 30.00 | 95.00 | 5.00 | 12.00 | 9.00 | 12.00 | 0.00 | 0.00 | 43.00 | 44.00 | 98.00 | 25.00 | 53.00 |
| has_cycle | 80.00 | 100.00 | 51.00 | 51.00 | 98.00 | 63.00 | 89.00 | 46.00 | 55.00 | 54.00 | 83.00 | 65.00 | 83.00 | 95.00 | 93.00 | 98.00 | 64.00 | 54.00 |
| dfs | 73.33 | 33.33 | 0.00 | 9.00 | 61.00 | 43.00 | 100.00 | 12.00 | 27.00 | 10.00 | 23.00 | 0.00 | 0.00 | 52.00 | 50.00 | 99.00 | 29.00 | 49.00 |
| minimum_spanning_tree | 38.10 | 42.86 | 5.00 | 8.00 | 62.00 | 17.00 | 81.00 | 17.00 | 15.00 | 14.00 | 46.00 | 0.00 | 0.00 | 65.00 | 65.00 | 66.00 | 28.00 | 39.00 |
| edge_existence | 100.00 | 100.00 | 60.00 | 80.00 | 100.00 | 97.00 | 100.00 | 73.00 | 96.00 | 82.00 | 100.00 | 52.00 | 56.00 | 98.00 | 97.00 | 66.00 | 98.00 | 100.00 |
| is_regular | 100.00 | 95.00 | 88.00 | 95.00 | 98.00 | 98.00 | 100.00 | 92.00 | 96.00 | 99.00 | 98.00 | 27.00 | 58.00 | 99.00 | 99.00 | 100.00 | 99.00 | 100.00 |
| degree | 95.45 | 100.00 | 26.00 | 58.00 | 94.00 | 93.00 | 95.00 | 72.00 | 77.00 | 79.00 | 96.00 | 22.00 | 0.00 | 88.00 | 82.00 | 99.00 | 94.00 | 82.00 |
| is_tournament | 100.00 | 88.89 | 47.00 | 75.00 | 99.00 | 99.00 | 99.00 | 80.00 | 86.00 | 87.00 | 100.00 | 0.00 | 58.00 | 100.00 | 100.00 | 99.00 | 99.00 | 94.00 |
| density | 68.18 | 90.91 | 36.00 | 33.00 | 17.00 | 38.00 | 92.00 | 42.00 | 38.00 | 40.00 | 81.00 | 0.00 | 0.00 | 12.00 | 15.00 | 97.00 | 51.00 | 51.00 |
| adamic_adar_index | 92.31 | 88.46 | 1.00 | 6.00 | 74.00 | 89.00 | 94.00 | 12.00 | 39.00 | 22.00 | 82.00 | 3.00 | 1.00 | 76.00 | 75.00 | 98.00 | 52.00 | 64.00 |
| clustering_coefficient | 72.22 | 94.44 | 13.00 | 31.00 | 71.00 | 56.00 | 82.00 | 25.00 | 44.00 | 36.00 | 65.00 | 6.00 | 10.00 | 67.00 | 66.00 | 88.00 | 49.00 | 69.00 |
| connected_component_number | 60.87 | 82.61 | 9.00 | 27.00 | 85.00 | 63.00 | 79.00 | 34.00 | 35.00 | 30.00 | 79.00 | 0.00 | 0.00 | 80.00 | 81.00 | 92.00 | 64.00 | 66.00 |
| bipartite_maximum_matching | 40.74 | 48.15 | 3.00 | 19.00 | 53.00 | 47.00 | 82.00 | 13.00 | 12.00 | 3.00 | 42.00 | 0.00 | 0.00 | 76.00 | 73.00 | 87.00 | 29.00 | 37.00 |
| local_connectivity | 96.15 | 100.00 | 57.00 | 62.00 | 93.00 | 86.00 | 90.00 | 53.00 | 74.00 | 79.00 | 82.00 | 53.00 | 66.00 | 97.00 | 98.00 | 96.00 | 77.00 | 69.00 |
| jaccard_coefficient | 100.00 | 100.00 | 23.00 | 48.00 | 81.00 | 84.00 | 95.00 | 44.00 | 77.00 | 70.00 | 95.00 | 3.00 | 5.00 | 78.00 | 76.00 | 100.00 | 87.00 | 93.00 |
| min_edge_covering | 10.53 | 31.58 | 1.00 | 2.00 | 23.00 | 17.00 | 51.00 | 0.00 | 1.00 | 1.00 | 37.00 | 0.00 | 0.00 | 18.00 | 17.00 | 50.00 | 16.00 | 15.00 |
| is_eularian | 86.36 | 95.45 | 78.00 | 81.00 | 95.00 | 89.00 | 98.00 | 82.00 | 81.00 | 89.00 | 92.00 | 33.00 | 59.00 | 93.00 | 93.00 | 97.00 | 90.00 | 80.00 |
| degree_centrality | 71.43 | 85.71 | 0.00 | 7.00 | 81.00 | 79.00 | 89.00 | 4.00 | 8.00 | 23.00 | 87.00 | 0.00 | 0.00 | 80.00 | 84.00 | 97.00 | 49.00 | 88.00 |
| is_bipartite | 68.00 | 92.00 | 49.00 | 39.00 | 92.00 | 55.00 | 79.00 | 53.00 | 52.00 | 43.00 | 76.00 | 51.00 | 67.00 | 93.00 | 90.00 | 92.00 | 62.00 | 67.00 |
| resource_allocation_index | 94.12 | 100.00 | 2.00 | 10.00 | 80.00 | 79.00 | 92.00 | 15.00 | 45.00 | 40.00 | 86.00 | 2.00 | 2.00 | 77.00 | 80.00 | 92.00 | 36.00 | 78.00 |
| max_weight_matching | 11.11 | 27.78 | 2.00 | 3.00 | 25.00 | 22.00 | 24.00 | 7.00 | 12.00 | 2.00 | 40.00 | 0.00 | 0.00 | 25.00 | 25.00 | 43.00 | 24.00 | 26.00 |
| closeness_centrality | 0.00 | 31.58 | 0.00 | 1.00 | 8.00 | 6.00 | 9.00 | 4.00 | 3.00 | 3.00 | 14.00 | 0.00 | 0.00 | 4.00 | 5.00 | 11.00 | 13.00 | 11.00 |
| traveling_salesman_problem | 36.84 | 89.47 | 8.00 | 24.00 | 29.00 | 40.00 | 43.00 | 17.00 | 41.00 | 41.00 | 62.00 | 3.00 | 1.00 | 25.00 | 20.00 | 51.00 | 47.00 | 43.00 |
| strongly_connected_number | 13.33 | 73.33 | 4.00 | 5.00 | 63.00 | 24.00 | 58.00 | 3.00 | 11.00 | 7.00 | 35.00 | 0.00 | 0.00 | 55.00 | 56.00 | 59.00 | 9.00 | 10.00 |
| shortest_path | 69.23 | 38.46 | 11.00 | 19.00 | 74.00 | 51.00 | 62.00 | 31.00 | 35.00 | 11.00 | 62.00 | 3.00 | 0.00 | 77.00 | 78.00 | 70.00 | 62.00 | 60.00 |
| center | 19.05 | 66.67 | 4.00 | 8.00 | 25.00 | 13.00 | 25.00 | 6.00 | 8.00 | 9.00 | 26.00 | 0.00 | 0.00 | 24.00 | 25.00 | 35.00 | 29.72 | 41.48 |
| diameter | 17.65 | 94.12 | 12.00 | 8.00 | 55.00 | 31.00 | 46.00 | 14.00 | 31.00 | 27.00 | 39.00 | 3.00 | 4.00 | 41.00 | 39.00 | 49.00 | 5.00 | 0.00 |
| barycenter | 7.69 | 69.23 | 9.00 | 15.00 | 56.00 | 26.00 | 39.00 | 20.00 | 22.00 | 11.11 | 29.00 | 1.01 | 1.01 | 49.00 | 50.00 | 47.00 | 53.71 | 47.61 |
| radius | 68.75 | 87.50 | 12.00 | 23.00 | 66.00 | 47.00 | 56.00 | 26.00 | 34.00 | 35.00 | 52.00 | 1.00 | 2.00 | 63.00 | 58.00 | 68.00 | 5.00 | 1.00 |
| topological_sort | 60.00 | 48.00 | 10.00 | 14.00 | 76.00 | 38.00 | 67.00 | 25.00 | 25.00 | 21.00 | 64.00 | 6.00 | 5.00 | 74.00 | 71.00 | 78.00 | 73.00 | 74.00 |
| periphery | 29.41 | 58.82 | 1.00 | 3.00 | 33.00 | 16.00 | 22.00 | 1.00 | 11.00 | 6.00 | 25.00 | 0.00 | 0.00 | 27.00 | 29.00 | 31.00 | 50.06 | 47.78 |
| betweenness_centrality | 18.18 | 50.00 | 4.00 | 4.00 | 38.00 | 30.00 | 39.00 | 24.00 | 1.00 | 5.00 | 6.00 | 1.00 | 2.00 | 38.00 | 37.00 | 39.00 | 7.00 | 4.00 |
| triangles | 35.29 | 58.82 | 13.00 | 4.00 | 54.00 | 48.00 | 67.00 | 12.00 | 30.00 | 21.00 | 54.00 | 0.00 | 0.00 | 42.00 | 40.00 | 79.00 | 43.00 | 55.00 |
| avg_neighbor_degree | 66.67 | 61.11 | 16.00 | 17.00 | 36.00 | 55.00 | 68.00 | 26.00 | 30.00 | 29.00 | 58.00 | 3.00 | 6.00 | 31.00 | 36.00 | 82.00 | 62.00 | 64.00 |
| harmonic_centrality | 7.69 | 84.62 | 2.00 | 3.00 | 17.00 | 15.00 | 19.00 | 3.00 | 5.00 | 8.00 | 37.00 | 1.00 | 2.00 | 9.00 | 7.00 | 30.00 | 7.00 | 22.00 |
| bridges | 0.00 | 9.09 | 1.00 | 0.00 | 44.00 | 9.00 | 16.00 | 0.00 | 3.00 | 1.00 | 5.00 | 0.00 | 0.00 | 42.00 | 40.00 | 23.00 | 28.57 | 29.92 |
| isomophic_mapping | 0.00 | 4.00 | 0.00 | 0.00 | 10.00 | 0.00 | 1.00 | 1.00 | 0.00 | 0.00 | 1.00 | 0.00 | 0.00 | 11.00 | 11.00 | 12.00 | 1.00 | 1.00 |
| global_efficiency | 4.76 | 71.43 | 0.00 | 1.00 | 10.00 | 3.00 | 1.00 | 0.00 | 2.00 | 2.00 | 11.00 | 0.00 | 0.00 | 4.00 | 4.00 | 2.00 | 1.00 | 3.00 |
| maximal_independent_set | 5.56 | 55.56 | 2.00 | 1.00 | 26.00 | 2.00 | 13.00 | 2.00 | 2.00 | 3.00 | 31.00 | 0.00 | 0.00 | 19.00 | 24.00 | 79.00 | 7.00 | 13.00 |
| maximum_flow | 0.00 | 80.95 | 5.00 | 2.00 | 8.00 | 10.00 | 7.00 | 3.00 | 6.00 | 1.10 | 12.00 | 3.30 | 5.49 | 9.00 | 7.00 | 9.00 | 4.00 | 10.00 |
| wiener_index | 0.00 | 73.68 | 0.00 | 1.00 | 14.00 | 6.00 | 8.00 | 0.00 | 4.00 | 4.00 | 22.00 | 0.00 | 0.00 | 6.00 | 3.00 | 13.00 | 7.00 | 7.00 |
| hamiltonian_path | 0.00 | 4.76 | 0.00 | 1.00 | 11.00 | 3.00 | 2.00 | 1.00 | 2.00 | 1.09 | 3.00 | 0.00 | 0.00 | 10.00 | 10.00 | 5.00 | 5.00 | 12.00 |
| min_vertex_cover | 13.04 | 60.87 | 1.00 | 3.00 | 22.00 | 8.00 | 21.00 | 3.00 | 9.00 | 6.00 | 21.00 | 0.00 | 0.00 | 16.00 | 18.00 | 42.00 | 10.00 | 21.00 |

## J  Discussion on Reward Weighting

In Section 5.3, we analyze the factor of data mixture by introducing a model G1-Hard-3B trained exclusively on *Hard* and *Challenging* tasks. We observe that G1-Hard-3B effectively improves performance on hard tasks, while on easier tasks still lags behind G1-3B (Table 24).

In this section, we further explore a *soft* data mixture strategy that scales the reward for each task according to its difficulty. In detail, we fix the scaling factor $s$ as 0.2, 0.4, 0.6, and 0.8 for *Easy*, *Medium*, *Hard* and *Challenging* tasks, respectively, and name the resulting model as G1-Soft-3B. As shown in Table 24, G1-Soft-3B achieves a balance between G1-3B and G1-Hard-3B. On easy tasks, G1-Soft-3B largely surpasses G1-Hard-3B and is on par with G1-3B which applies uniform scaling across all tasks. For hard tasks, G1-Soft-3B outperforms G1-3B (*e.g.*, 11.71% v.s 7.57% for *Challenging* tasks), but there is still a gap to G1-Hard-3B. The results show the soft scaling method take effects, but the RL optimization remains dominated by easy tasks. This suggests that further reducing the reward scaling factor for easy tasks or a dynamic weighting strategy could be beneficial—a direction we leave for future work.

Table 24: Test accuracy (%) on our benchmark. ⋆ denotes the tasks are excluded in model training. G1-Hard-3B is only RL-trained on Hard and Challenging tasks. G1-Soft-3B is trained on all tasks but with different reward scaling factors based on the task difficulty.

| Category | Model | Easy | Medium | Hard | Challenging | Average |
|---|---|---|---|---|---|---|
| **Base Model** | Qwen2.5-3B-Instruct | 45.71 | 30.18 | 9.44 | 1.29 | 22.72 |
| **Ours** | Direct-SFT-3B | 74.43 | 75.27 | 43.69 | 14.43 | 53.78 |
| | G1-3B | 94.86 | **84.64** | 41.25 | 7.57 | 59.76 |
| | G1-Hard-3B | 69.36⋆ | 70.64⋆ | **48.50** | **17.43** | 53.30 |
| | G1-Soft-3B | **96.07** | 83.55 | 40.88 | 11.71 | **60.38** |

## K  Detailed Description of Erdős

### K.1  Comparing Erdős with Other Graph Reasoning Benchmarks for LLMs

There is a growing interest in evaluating LLMs' graph reasoning abilities. NLGraph [41] evaluate LLMs on graph-theoretic tasks and discover preliminary yet brittle reasoning abilities in the face of spurious correlations and large graphs. Later, GraphArena [38] and GraCoRe [55] include a broader task coverage and recently released LLMs, finding that even OpenAI o1-mini struggles a lot with complex tasks. Moreover, GraphEval2000 [46] and ProGraph [23] emphasize code-oriented problem solving using library-based prompts, and GraphOmni [48] unify varying graph types, encodings, and prompt styles for a comprehensive evaluation. Overall, these benchmarks suggest that LLMs overall demonstrate moderate success on simple tasks but struggle with abstraction, generalization, and larger or more complex graph instances. Nevertheless, these datasets are either too small (e.g., thousands of examples) or not diverse enough (e.g., 8 tasks in NLGraph) for training general-purpose graph reasoners, which motivates the design of Erdős. We show the detailed comparison of existing graph reasoning benchmarks for LLM with our Erdősin Table 25.

Table 25: Comparison of existing graph-theoretic reasoning benchmarks for LLM with our Erdős.

| Benchmark | #Tasks | # Q-A Samples | Graph Types | Node Size |
|---|---|---|---|---|
| NLGraph [41] | 8 | 5,902 | Synthetic | 5 to 35 |
| GraphWiz [3] | 9 | 3,600 | Synthetic | 2 to 100 |
| GraphArena [38] | 10 | 10,000 | Real-world | 4 to 50 |
| GraCoRe [55] | 19 | 5,140 | Synthetic & Real-world | 8 to 30 |
| GraphOmni [48] | 6 | 241,726 | Synthetic | 5 to 30 |
| Erdős(**ours**) | 50 | 100,000 | Real-world | 5 to 35 |

## K.2 Full list of tasks in `Erdős`

Table 26: Benchmark exmaples

| Task | Prompt | Answer |
|------|--------|--------|
| **adamic adar index** | The task is to determine the Adamic-Adar index of two nodes in a graph.
The Adamic-Adar index is the sum of the inverse logarithm of the degrees of the common neighbors of the two nodes.
The input graph is guaranteed to be undirected.
Here is an undirected graph containing nodes from 1 to 9. The edges are: (1, 5), (1, 4), (1, 8), (1, 2), (1, 3), (1, 7), (5, 2), (5, 3), (5, 4), (5, 9), (5, 6), (4, 8), (4, 9), (4, 7), (8, 2), (8, 3), (8, 6), (8, 7), (8, 9), (2, 3), (2, 7), (2, 6), (3, 9), (3, 7), (7, 6), (7, 9).
Question: What is the Adamic-Adar index between node 4 and node 6?
You need to format your answer as a float number. | 1.5859 |
| **avg neighbor degree** | The task is to determine the average degree of the neighbors of a node in the graph.
Here is an undirected graph containing nodes from 1 to 8. The edges are: (1, 7), (1, 8), (1, 4), (7, 8), (8, 5), (2, 3), (2, 6), (3, 5).
Question: What is the average neighbor degree of node 2 in the graph?
You need to format your answer as a float number. | 1.5 |
| **barycenter** | The task is to determine the barycenter of a graph.
The barycenter of a graph is also called the median. It includes the node that minimizes the sum of shortest path lengths to all other nodes.
The input graph is guaranteed to be connected.
Here is an undirected graph containing nodes from 1 to 7. The edges are: (1, 2), (1, 6), (1, 5), (1, 7), (1, 4), (2, 6), (2, 5), (2, 7), (2, 4), (6, 4), (6, 5), (6, 7), (7, 3), (7, 4).
Question: What is the barycenter of the graph?
You need to format your answer as a list of nodes in ascending order, e.g., [node-1, node-2, ..., node-n]. | [1, 2, 6, 7] |
| **betweenness centrality** | The task is to determine the betweenness centrality of a node in the graph.
Betweenness centrality of a node *u* is the sum of the fraction of all-pairs shortest paths that pass through *u*.
Here is an undirected graph containing nodes from 1 to 9. The edges are: (1, 6), (1, 4), (1, 8), (1, 9), (6, 2), (6, 7), (4, 7), (4, 5), (8, 3), (8, 5), (8, 7), (9, 3), (9, 5), (2, 7).
Question: What is the betweenness centrality of node 5 in the graph?
You need to format your answer as a float number. | 0.0679 |
| **bfs** | The task is to determine the breadth-first search (BFS) traversal order given a starting node.
Stop when the BFS cannot be continued.
Here is an undirected graph containing nodes from 1 to 7. The edges are: (1, 2), (1, 5), (2, 3), (2, 4), (5, 3), (5, 4), (3, 4), (4, 7), (7, 6).
Question: What is the breadth-first search (BFS) traversal order for the starting node 1?
You need to format your answer as a list of edges, e.g., [(u1, v1), (u2, v2), ..., (un, vn)]. | [(1, 2), (1, 5), (2, 3), (2, 4), (4, 7), (7, 6)] |

Continuing table 26

| Task | Prompt | Answer |
|---|---|---|
| **bipartite maximum matching** | The task is to determine the maximal matching in a bipartite graph.
The input graph is guaranteed to be a bipartite graph.
Here is an undirected graph containing nodes from 1 to 4. The edges are: (1, 3), (1, 4), (2, 3), (2, 4).
Question: What is the bipartite maximal matching of the bipartite graph?
You need to format your answer as a list of edges in ascending dictionary order, e.g., [(u1, v1), (u2, v2), ..., (un, vn)]. | [(1, 3), (2, 4)] |
| **bridges** | The task is to find all bridges of a graph.
A bridge is an edge in a graph whose removal increases the number of connected components.
The input graph is guaranteed to be undirected.
Here is an undirected graph containing nodes from 1 to 5. The edges are: (1, 2), (1, 3), (1, 4), (2, 3), (2, 4), (2, 5), (3, 4), (3, 5).
Question: What are the bridges of the graph?
You need to format your answer as a list of edges in ascending dictionary order, e.g., [(u1, v1), (u2, v2), ..., (un, vn)]. | [] |
| **center** | The task is to determine the center of a graph.
The center of a graph includes the node that minimizes the maximum distance to any other nodes in the graph.
The input graph is guaranteed to be connected.
Here is an undirected graph containing nodes from 1 to 6. The edges are: (1, 5), (5, 2), (2, 6), (6, 4), (3, 4).
Question: What is the center of the graph?
You need to format your answer as a list of nodes in ascending order, e.g., [node-1, node-2, ..., node-n]. | [2, 6] |
| **closeness centrality** | The task is to determine the closeness centrality of a node in the graph.
For a node *u*, closeness centrality is the reciprocal of the average shortest path distance to *u* over all *n-1* reachable nodes. For directed graphs, it computes the incoming distance to *u*.
Here is an undirected graph containing nodes from 1 to 8. The edges are: (1, 3), (3, 6), (2, 8), (2, 6), (8, 6), (8, 7), (4, 7), (7, 5).
Question: What is the closeness centrality of node 2 in the graph?
You need to format your answer as a float number. | 0.4667 |
| **clustering coefficient** | The task is to compute the clustering coefficient for a given node.
For unweighted graphs, the clustering of a node is the fraction of possible triangles through that node that exist.
Here is an undirected graph containing nodes from 1 to 7. The edges are: (1, 4), (1, 5), (1, 3), (4, 2), (4, 3), (4, 5), (4, 6), (4, 7), (5, 2), (5, 3), (5, 6), (5, 7), (2, 6), (2, 7), (6, 7).
Question: What is the clustering coefficient of node 6?
You need to format your answer as a float number. | 1.0 |
| **common neighbor** | The task is to determine common neighbors between two nodes in the graph.
The input graph is guaranteed to be undirected.
Here is an undirected graph containing nodes from 1 to 7. The edges are: (1, 7), (1, 6), (1, 4), (1, 5), (7, 2), (7, 3), (6, 2), (4, 3), (5, 3).
Question: What are the common neighbors between node 2 and node 3?
You need to format your answer as a list of nodes in ascending order, e.g., [node-1, node-2, ..., node-n]. | [7] |

| Task | Prompt | Answer |
|------|--------|--------|
| **connected component number** | The task is to determine the number of connected components in an undirected graph.
A connected component is a subgraph where any two nodes are connected to each other by paths.
Here is an undirected graph containing nodes from 1 to 10. The edges are: (1, 4), (1, 7), (1, 5), (1, 9), (1, 10), (1, 6), (1, 2), (4, 2), (4, 3), (4, 8), (4, 5), (4, 9), (4, 10), (7, 2), (7, 3), (7, 5), (7, 6), (7, 8), (7, 9), (5, 2), (5, 3), (5, 8), (5, 9), (5, 10), (9, 2), (9, 3), (9, 6), (9, 8), (9, 10), (10, 2), (10, 3), (10, 8), (6, 2), (6, 3), (6, 8), (2, 8), (2, 3).
Question: How many connected components are there in the graph?
Your answer should be an integer. | 1 |
| **degree** | The task is to determine the degree of a node in the graph.
For the undirected graph, you should count the edge between two nodes only once.
Here is an undirected graph containing nodes from 1 to 6. The edges are: (1, 6), (6, 5), (2, 3), (2, 4), (3, 5).
Question: What is the degree of node 6 in the graph?
Your answer should be an integer. | 2 |
| **degree centrality** | The task is to determine the degree centrality of a node in the graph.
Degree centrality for a node is the fraction of nodes it is connected to.
Here is an undirected graph containing nodes from 1 to 7. The edges are: (1, 2), (1, 4), (1, 5), (2, 3), (2, 4), (2, 5), (2, 6), (4, 3), (4, 5), (4, 7), (5, 3).
Question: What is the degree centrality of node 3 in the graph?
You need to format your answer as a float number. | 0.5 |
| **density** | The task is to determine the density of the graph.
Density is defined as the ratio of the number of edges in the graph to the number of possible edges.
Here is an undirected graph containing nodes from 1 to 5. The edges are: (1, 2), (1, 3), (2, 3), (2, 4), (2, 5), (3, 4), (4, 5).
Question: What is the density of the graph?
You need to format your answer as a float number. | 0.7 |
| **dfs** | The task is to determine the depth-first search (DFS) traversal order given a starting node.
Stop when the DFS cannot be continued.
Here is an undirected graph containing nodes from 1 to 9. The edges are: (1, 2), (1, 3), (1, 6), (3, 9), (4, 8), (4, 5), (8, 7).
Question: What is the depth-first search (DFS) traversal order for the starting node 1?
You need to format your answer as a list of edges, e.g., [(u1, v1), (u2, v2), ..., (un, vn)]. | [(1, 2), (1, 3), (3, 9), (1, 6)] |
| **diameter** | The task is to determine the diameter of a graph.
The diameter of a graph is the longest shortest path between any two nodes in the graph.
The input graph is guaranteed to be connected.
Here is an undirected graph containing nodes from 1 to 7. The edges are: (1, 5), (1, 7), (1, 4), (5, 6), (2, 6), (2, 3).
Question: What is the diameter of the graph?
You need to format your answer as a float number. | 5 |

| Task | Prompt | Answer |
|---|---|---|
| **dominating set** | The task is to determine the dominating set of a graph. A dominating set is a subset of nodes such that every node in the graph is either in the set or adjacent to a node in the set. For directed graphs, any node not in the dominating set must be a successor of a node within the set. Here is an undirected graph containing nodes from 1 to 7. The edges are: (1, 2), (1, 5), (1, 6), (1, 7), (2, 3), (2, 4), (5, 6), (7, 3), (7, 4). Question: What is the dominating set of the graph? You need to format your answer as a list of nodes in ascending order, e.g., [node-1, node-2, ..., node-n]. | [1, 3, 4] |
| **edge existence** | The task is to determine if there is an edge connecting two nodes. For an undirected graph, determine if there is an edge between nodes *u* and *v*. For a directed graph, determine if there is an edge from *u* to *v*. Here is an undirected graph containing nodes from 1 to 8. The edges are: (1, 2), (1, 6), (3, 8), (3, 4), (8, 4), (8, 5), (8, 7), (4, 7), (4, 5), (7, 5). Question: Is there an edge between node 5 and node 3? Your answer should be Yes or No. | No |
| **edge number** | The task is to determine the number of edges in the graph. For the undirected graph, you should count the edge between two nodes only once. Here is an undirected graph containing nodes from 1 to 10. The edges are: (1, 10), (1, 8), (10, 7), (8, 6), (2, 5), (2, 4), (2, 6), (5, 4), (5, 9), (4, 3), (4, 9), (3, 7). Question: How many edges are there in the graph? Your answer should be an integer. | 12 |
| **global efficiency** | The task is to determine the global efficiency of a graph. Global efficiency is the average efficiency of all pairs of nodes. The efficiency of a pair of nodes is the multiplicative inverse of the shortest path distance between the nodes. The input graph is guaranteed to be undirected. Here is an undirected graph containing nodes from 1 to 7. The edges are: (1, 5), (1, 4), (5, 2), (2, 7), (7, 3), (3, 6). Question: What is the global efficiency of the graph? You need to format your answer as a float number. | 0.5310 |
| **hamiltonian path** | The task is to return a Hamiltonian path in a directed graph. A Hamiltonian path is a path in a directed graph that visits each vertex exactly once. The input graph is guaranteed to be directed and tournable. Here is a directed graph containing nodes from 1 to 8. The edges are: (2, 1), (2, 4), (2, 5), (2, 6), (2, 7), (1, 3), (1, 4), (1, 7), (3, 2), (3, 7), (3, 8), (4, 3), (4, 5), (4, 7), (5, 1), (5, 3), (5, 8), (6, 1), (6, 3), (6, 4), (6, 5), (7, 5), (7, 6), (8, 1), (8, 2), (8, 4), (8, 6), (8, 7). Question: Return a Hamiltonian path in the graph. You need to format your answer as a list of nodes, e.g., [node-1, node-2, ..., node-n]. | [2, 1, 4, 5, 3, 8, 7, 6] |

| Task | Prompt | Answer |
|------|--------|--------|
| **harmonic centrality** | The task is to determine the harmonic centrality of a node in the graph.
Harmonic centrality of a node *u* is the sum of the reciprocal of the shortest path distances from all other nodes to u.
Here is a directed graph containing nodes from 1 to 8. The edges are: (6, 2), (6, 1), (6, 4), (6, 5), (6, 3), (7, 8).
Question: What is the harmonic centrality of node 3 in the graph?
You need to format your answer as a float number. | 1.0 |
| **has cycle** | The task is to determine if the graph has a cycle.
Here is an undirected graph containing nodes from 1 to 9. The edges are: (1, 2), (1, 4), (1, 5), (2, 4), (2, 5), (4, 9), (5, 3), (3, 6), (3, 8), (6, 8), (9, 7).
Question: Does the graph have a cycle?
Your answer should be Yes or No. | Yes |
| **is bipartite** | The task is to determine if the graph is bipartite.
A bipartite graph is a graph whose nodes can be divided into two disjoint sets such that no two graph vertices within the same set are adjacent.
Here is an undirected graph containing nodes from 1 to 6. The edges are: (1, 4), (4, 3), (2, 5), (2, 3), (5, 6), (3, 6).
Question: Is the graph bipartite?
Your answer should be Yes or No. | Yes |
| **is eularian** | The task is to determine if the graph is Eulerian.
An Eulerian graph is a graph that contains an Eulerian circuit, which is a cycle that visits every edge exactly once.
Here is an undirected graph containing nodes from 1 to 6. The edges are: (1, 5), (1, 3), (1, 2), (1, 4), (5, 2), (3, 2), (3, 4), (3, 6), (2, 4), (4, 6).
Question: Is the graph Eulerian?
Your answer should be Yes or No. | Yes |
| **is regular** | The task is to determine if the graph is regular.
A regular graph is a graph where every node has the same degree.
Here is an undirected graph containing nodes from 1 to 10. The edges are: (1, 5), (1, 7), (1, 10), (5, 2), (5, 10), (7, 8), (7, 10), (3, 9), (3, 8), (3, 4), (9, 4), (4, 6).
Question: Is the graph regular?
Your answer should be Yes or No. | No |
| **is tournament** | The task is to determine if the graph is a tournament.
A tournament is a directed graph where every pair of nodes is connected by a single directed edge.
The input graph is guaranteed to be directed.
Here is a directed graph containing nodes from 1 to 10. The edges are: (1, 2), (2, 1), (2, 4), (4, 2), (4, 3), (3, 1), (5, 2), (5, 4), (6, 2), (6, 5), (7, 8), (8, 6), (9, 7), (10, 7).
Question: Is the graph a tournament?
Your answer should be Yes or No. | No |

| Task | Prompt | Answer |
|------|--------|--------|
| **isomophic mapping** | Given a pair of isomorphic graphs, determine the node correspondence between the two graphs. The first graph is: G describes an undirected graph among 0, 1, 2, 3, 4, 5, and 6. In this graph: Node 0 is connected to nodes 6, 3, 4. Node 1 is connected to nodes 4, 5, 6. Node 2 is connected to nodes 3, 4. Node 3 is connected to nodes 0, 2, 5. Node 4 is connected to nodes 0, 1, 2. Node 5 is connected to nodes 1, 3. Node 6 is connected to nodes 0, 1. The second graph is: G describes an undirected graph among 102, 106, 105, 101, 103, 100, and 104. In this graph: Node 100 is connected to nodes 106, 101. Node 101 is connected to nodes 102, 105, 100. Node 102 is connected to nodes 104, 101, 103. Node 103 is connected to nodes 102, 106, 105. Node 104 is connected to nodes 102, 106. Node 105 is connected to nodes 101, 103. Node 106 is connected to nodes 103, 100, 104. Provide a node matching dictionary such as {Graph1 #Node1: Graph2 #Node1, Graph1 #Node2: Graph2 #Node2, ...} | {0: 102, 3: 101, 2: 105, 4: 103, 1: 106, 5: 100, 6: 104} |
| **jaccard coefficient** | The task is to determine the Jaccard coefficient of two nodes in a graph. The Jaccard coefficient is the size of the intersection divided by the size of the union of the neighbors of the two nodes. The input graph is guaranteed to be undirected. Here is an undirected graph containing nodes from 1 to 5. The edges are: (1, 2), (1, 3), (2, 5), (2, 3), (3, 5), (5, 4). Question: What is the Jaccard coefficient between node 2 and node 4? You need to format your answer as a float number. | 0.3333 |
| **local connectivity** | The task is to determine the local connectivity of two nodes in the graph. Local connectivity is whether there exists at least one path between the two nodes. Here is a directed graph containing nodes from 1 to 7. The edges are: (1, 7), (7, 6), (3, 1), (4, 3), (5, 4), (6, 2). Question: What is the local connectivity between node 7 and node 4 in the graph? Your answer should be Yes or No. | No |
| **max weight matching** | The task is to determine the maximum weight matching of a graph. A matching is a set of edges without common vertices. A maximal matching cannot add more edges and still be a matching. The weight of a matching is the sum of the weights of its edges. If not specified, all edges have equal edge weights. The input graph is guaranteed to be undirected. Here is an undirected graph containing nodes from 1 to 7. The edges are: (1, 7), (7, 5), (2, 4), (2, 5), (4, 3), (3, 6). Question: What is the maximum weight matching of the graph? You need to format your answer as a list of edges in ascending dictionary order, e.g., [(u1, v1), (u2, v2), ..., (un, vn)]. | [(2, 4), (5, 7), (6, 3)] |

| Task | Prompt | Answer |
|---|---|---|
| maximal independent set | The task is to determine the maximal independent set guaranteed to contain a given node in the graph. An independent set is a set of nodes such that the subgraph induced by these nodes contains no edges. A maximal independent set is an independent set such that it is not possible to add a new node and still get an independent set. The input graph is guaranteed to be undirected. Here is an undirected graph containing nodes from 1 to 6. The edges are: (1, 2), (1, 6), (1, 3), (2, 3), (2, 4), (2, 5), (3, 5), (4, 5). Question: What is the maximal independent set that includes node 4 of the graph? You need to format your answer as a list of nodes in ascending order, e.g., [node-1, node-2, ..., node-n]. | [3, 4, 6] |
| maximum flow | The task is to determine the value of the maximum flow for the given source node and sink node. The maximum flow is the greatest amount of flow that can be sent from the source to the sink without violating capacity constraints. Here is a directed graph containing nodes from 1 to 5. The edges are: (2, 5, 8), (3, 1, 9), (3, 5, 3), (4, 2, 4). (u ,v, w) denotes the edge from node *u* to node *v* has a capacity of *w*. Question: What is the value of the maximum flow from node 3 to node 2? You need to format your answer as a float number. | 0.0 |
| min edge covering | The task is to determine the minimum edge covering of a graph. An edge cover is a set of edges such that every vertex in the graph is incident to at least one edge in the set. The minimum edge cover is the edge cover with the smallest number of edges. The input graph is guaranteed to be undirected. Here is an undirected graph containing nodes from 1 to 9. The edges are: (1, 2), (1, 3), (1, 4), (2, 3), (2, 4), (2, 5), (3, 4), (3, 6), (3, 7), (3, 8), (3, 5), (4, 7), (4, 8), (5, 6), (5, 7), (6, 7), (6, 9), (7, 9). Question: What is the minimum edge covering of the graph? You need to format your answer as a list of edges in ascending dictionary order, e.g., [(u1, v1), (u2, v2), ..., (un, vn)]. | [(2, 1), (5, 2), (7, 4), (8, 3), (9, 6)] |
| min vertex cover | The task is to determine the minimum vertex cover of a graph. A vertex cover is a set of nodes such that every edge in the graph is incident to at least one node in the set. Here is an undirected graph containing nodes from 1 to 5. The edges are: (1, 2), (2, 3), (3, 5), (5, 4). Question: What is the minimum vertex cover of the graph? You need to format your answer as a list of nodes in ascending order, e.g., [node-1, node-2, ..., node-n]. | [2, 5] |
| minimum spanning tree | The task is to determine the minimum spanning tree of a graph. A minimum spanning tree is a subset of the edges that connects all vertices in the graph with the minimum possible total edge weight. If not specified, all edges have equal edge weights. The input graph is guaranteed to be undirected and connected. Here is an undirected graph containing nodes from 1 to 9. The edges are: (1, 2), (1, 8), (1, 5), (1, 6), (1, 4), (1, 7), (1, 9), (2, 5), (2, 6), (2, 4), (2, 7), (2, 3), (8, 3), (8, 4), (8, 6), (8, 7), (5, 3), (5, 4), (5, 6), (5, 7), (5, 9), (6, 3), (6, 4), (6, 7), (6, 9), (4, 3), (4, 7), (4, 9), (7, 9), (9, 3). Question: What is the minimum spanning tree of the graph? You need to format your answer as a list of edges in ascending dictionary order, e.g., [(u1, v1), (u2, v2), ..., (un, vn)]. | [(1, 2), (1, 4), (1, 5), (1, 6), (1, 7), (1, 8), (1, 9), (2, 3)] |

Continuing table 26

| Task | Prompt | Answer |
|---|---|---|
| **neighbor** | The task is to determine the neighbors of a node in the graph. For directed graph, you should return the successors of the node.
Here is an undirected graph containing nodes from 1 to 10. The edges are: (1, 3), (1, 9), (1, 6), (1, 7), (3, 2), (3, 8), (3, 9), (6, 7), (2, 10), (10, 8), (4, 5).
Question: What are the neighbors of node 2 in the graph?
You need to format your answer as a list of nodes in ascending order, e.g., [node-1, node-2, ..., node-n]. | [3, 10] |
| **node number** | The task is to determine the number of nodes in the graph.
Here is an undirected graph containing nodes from 1 to 10. The edges are: (1, 10), (1, 3), (10, 6), (10, 8), (3, 7), (3, 4), (2, 7), (2, 5), (2, 9), (5, 9), (5, 8), (9, 4), (8, 6).
Question: How many nodes are there in the graph?
Your answer should be an integer. | 10 |
| **periphery** | The task is to determine the periphery of a graph.
The periphery of a graph is the set of nodes with the maximum eccentricity. The eccentricity of a node is the maximum distance from this node to all other nodes in the graph.
The input graph is guaranteed to be connected.
Here is an undirected graph containing nodes from 1 to 6. The edges are: (1, 3), (3, 2), (3, 4), (3, 5), (3, 6).
Question: What is the periphery of the graph?
You need to format your answer as a list of nodes in ascending order, e.g., [node-1, node-2, ..., node-n]. | [1, 2, 4, 5, 6] |
| **radius** | The task is to determine the radius of a graph.
The radius of a graph is the minimum eccentricity of any node in the graph. The eccentricity of a node is the maximum distance from this node to all other nodes in the graph.
The input graph is guaranteed to be connected.
Here is an undirected graph containing nodes from 1 to 5. The edges are: (1, 2), (2, 3), (3, 4), (3, 5), (4, 5).
Question: What is the radius of the graph?
You need to format your answer as a float number. | 2 |
| **resource allocation index** | The task is to determine the resource allocation index of two nodes in a graph.
The resource allocation index of two nodes is the sum of the inverse of the degrees of the common neighbors of the two nodes.
The input graph is guaranteed to be undirected.
Here is an undirected graph containing nodes from 1 to 5. The edges are: (1, 2), (1, 3), (2, 3), (3, 4), (3, 5), (4, 5).
Question: What is the resource allocation index between node 1 and node 4?
You need to format your answer as a float number. | 0.25 |
| **shortest path** | The task is to determine the shortest path between two nodes.
The input nodes are guaranteed to be connected.
Here is an undirected graph containing nodes from 1 to 6. The edges are: (1, 2), (1, 3), (2, 4), (2, 3), (2, 5), (3, 4), (3, 5), (4, 6).
Question: What is the shortest path between node 1 and node 6?
You need to format your answer as a list of nodes, e.g., [node-1, node-2, ..., node-n]. | [1, 2, 4, 6] |

| Task | Prompt | Answer |
|------|--------|--------|
| **strongly connected number** | The task is to determine the number of strongly connected components in a directed graph.
A strongly connected component is a maximal subgraph where every node is reachable from every other node.
Here is a directed graph containing nodes from 1 to 6. The edges are: (2, 5), (5, 1), (3, 4), (6, 2).
Question: How many strongly connected components are there in the graph?
Your answer should be an integer. | 6 |
| **topological sort** | The task is to determine the topological sort of a directed acyclic graph (DAG).
Here is a directed graph containing nodes from 1 to 6. The edges are: (1, 6), (1, 5), (1, 4), (1, 3), (1, 2).
Question: What is the topological sort of the directed acyclic graph (DAG)?
You need to format your answer as a list of nodes, e.g., [node-1, node-2, ..., node-n]. | [1, 6, 5, 4, 3, 2] |
| **traveling salesman problem** | The task is to determine the minimal cost of the Traveling Salesman Problem (TSP).
The Traveling Salesman Problem asks for the shortest possible route that visits each vertex exactly once and returns to the starting vertex.
The input graph is guaranteed to be a complete graph.
Here is an undirected graph containing nodes from 1 to 8. The edges are: (1, 2, 9), (1, 3, 3), (1, 4, 6), (1, 5, 8), (1, 6, 7), (1, 7, 4), (1, 8, 9), (2, 3, 10), (2, 4, 11), (2, 5, 5), (2, 6, 11), (2, 7, 1), (2, 8, 9), (3, 4, 11), (3, 5, 1), (3, 6, 9), (3, 7, 2), (3, 8, 9), (4, 5, 8), (4, 6, 3), (4, 7, 4), (4, 8, 8), (5, 6, 3), (5, 7, 3), (5, 8, 10), (6, 7, 8), (6, 8, 1), (7, 8, 10). (u ,v, w) denotes the edge from node *u* to node *v* has a weight of *w*.
Question: What is the minimal cost of the Traveling Salesman Problem on the graph?
You need to format your answer as a float number. | 27.0 |
| **triangles** | The task is to find the number of triangles that include a specific node as one vertex.
A triangle is a set of three nodes that are all connected to each other.
The input graph is guaranteed to be undirected.
Here is an undirected graph containing nodes from 1 to 8. The edges are: (1, 2), (1, 3), (1, 4), (1, 5), (1, 6), (1, 7), (1, 8), (2, 3), (2, 4), (2, 5), (2, 6), (2, 7), (2, 8), (3, 4), (3, 5), (3, 6), (3, 7), (3, 8), (4, 5), (4, 6), (4, 7), (4, 8), (5, 6), (5, 7), (5, 8), (6, 7), (6, 8), (7, 8).
Question: How many triangles include node 1 in the graph?
Your answer should be an integer. | 21 |

| Task | Prompt | Answer |
|---|---|---|
| **weighted minimum spanning tree** | The task is to determine the minimum spanning tree of a weighted graph.
A minimum spanning tree is a subset of the edges that connects all vertices in the graph with the minimum possible total edge weights. If not specified, all edges have equal edge weights.
The input graph is guaranteed to be undirected and connected.
Here is an undirected graph containing nodes from 1 to 5. The edges are: (1, 4, 5), (2, 4, 11), (2, 3, 10), (3, 4, 2), (3, 5, 2). (u ,v, w) denotes the edge from node *u* to node *v* has a weight of *w*.
Question: What is the minimum spanning tree of the weighted graph?
You need to format your answer as a list of edges in ascending dictionary order, e.g., [(u1, v1), (u2, v2), ..., (un, vn)]. | [(1, 4), (2, 3), (3, 4), (3, 5)] |
| **weighted shortest path** | The task is to determine the shortest path between two nodes of a weighted graph.
The input nodes are guaranteed to be connected.
Here is a directed graph containing nodes from 1 to 8. The edges are: (1, 2, 5), (1, 4, 3), (1, 7, 9), (2, 3, 10), (2, 4, 10), (3, 1, 11), (3, 4, 2), (3, 5, 6), (4, 1, 1), (4, 2, 4), (4, 6, 8), (4, 8, 2), (5, 1, 7), (5, 2, 11), (5, 6, 2), (5, 7, 5), (5, 8, 11), (6, 1, 7), (6, 2, 11), (6, 3, 4), (6, 5, 1), (6, 8, 11), (7, 1, 3), (7, 2, 8), (7, 4, 7), (7, 6, 6), (7, 8, 3), (8, 1, 11), (8, 2, 7), (8, 4, 5), (8, 7, 5). (u ,v, w) denotes the edge from node *u* to node *v* has a weight of *w*.
Question: What is the shortest path between node 1 and node 5?
You need to format your answer as a list of nodes, e.g., [node-1, node-2, ..., node-n]. | [1, 4, 6, 5] |
| **wiener index** | The task is to determine the Wiener index of a connected graph. The Wiener index of a graph is the sum of the shortest-path distances between each pair of reachable nodes. For pairs of nodes in undirected graphs, only one orientation of the pair is counted.
In the input graph, all node pairs are guaranteed to be reachable.
Here is an undirected graph containing nodes from 1 to 5. The edges are: (1, 2), (1, 4), (2, 3), (4, 5), (3, 5).
Question: What is the Wiener index of the graph?
You need to format your answer as a float number. | 15.0 |

