# OpenReview forum: "$\texttt{G1}$: Teaching LLMs to Reason on Graphs with Reinforcement Learning"
_NeurIPS.cc/2025/Conference — NeurIPS 2025 poster_

### Official Review · Reviewer_2Hnx · 2025-07-01

**Clarity:** 4
**Significance:** 4
**Originality:** 3
**Rating:** 6
**Confidence:** 4

**Summary:**

The paper provides a new LLM approach training with reinforcement learning to reason on graph tasks. The authors conduct extensive experiments on multiple datasets including synthetic and real-world datasets, comparing to other state-of-the-art comparable LLMs. The method achieves better performance on various graph tasks compared to baselines even though it uses less number of parameters.

**Questions:**

1. How can this model scale with it is being trained with very large number of parameters? Can you provide the running times of the current model? Can you provide some insights on what would happen if the method has much more parameters (e.g., 100B)?
2. What are the future directions of this particular research? Can you provide a section that discusses limitations and future directions?
3. Can you provide the robustness/correctness of the method by running the same questions multiple times on some tasks?

**Ethical Concerns:**

["NO or VERY MINOR ethics concerns only"]

**Final Justification:**

The paper quality is very strong and the authors made a great rebuttal, addressing all of my concerns and comments, regarding the following points: (1) Scalability of the method, (2) Experimental setting, (3) Future directions, (4) Limitations and societal aspect. That's why I'm increasing my score to strong accept.

**Limitations:**

Limitations and societal impacts are not clear. I suggest authors to make them clear. They mentioned that the limitations are in experiment section, which I don't see them. Societal impacts are supposed to be in appendix, again I don't see them. Can authors make them clearer, maybe in a separate section? They can be included with future directions that I already suggested to add.

**Quality:**

4

**Strengths And Weaknesses:**

Strengths:
1. The paper is very well written and structured. It is easy to follow and read.
2. The paper contributes to the research community by promising to share the code and all tasks that are used to train.
3. The paper conducts many experiments on multiple different graph tasks, and each tasks and experiments are explained really well.
4. The paper provides multiple versions of their methods which are serving as ablation study. They compare each method in training analysis and show the differences between the methods.

Weakness:
1. The method is only compared with methods with small parameters (there are some unknown methods and they may achieve better performance in some cases.)
2. The method uses only 7B parameters. How would the method scale if we want to build the larger model?
3. The paper does not really discuss the future directions, which is needed for the research paper.
4. All the experiments are run only once. The authors mentioned that randomness is negligible. LLMs are known to give different answers based on the questions. So ML practice suggests that multiple run is needed.

---

> ### Author Rebuttal · Authors · 2025-07-31
>
> We are grateful for Reviewer 2Hnx's positive feedback on our paper's clear writing and structure, the thoroughness of our experiments and ablation studies, and our contribution to the research community. We will address the remaining concerns in the discussion below.
>
> -----
>
> **Q1.** The method is only compared with methods with small parameters (there are some unknown methods and they may achieve better performance in some cases.)
>
> **A1.** In **Table 2**, we provide comparisons with several large-scale models, including Qwen2.5-72B-Instruct and Llama3.1-70B-Instruct (72B parameters). Notably, our G1-3B model achieves competitive performance against these 72B-parameter models while using only about 1/20 of their parameters. This demonstrates the effectiveness of our approach in achieving strong performance at a much smaller scale.
>
> ---
>
> **Q2.** The method uses only 7B parameters. How would the method scale if we want to build a larger model? How can this model scale with it is being trained with very large number of parameters? Can you provide the running times of the current model? Can you provide some insights on what would happen if the method has much more parameters (e.g., 100B)?
>
> **A2.** For **current models**, G1-7B was trained in approximately 40 hours on 8×A100 GPUs, while G1-3B required around 35 hours on 4×A100 GPUs. We will include these details in the paper. To demonstrate scalability, we further trained **G1-Zero-32B** from Qwen2.5-32B-Instruct using the same method.  G1-Zero-32B achieves significant average improvement on Erdős (_Table R1_) while maintaining strong math performance (_Table R2_), The training is completed in 40h on 32×A100 GPUs. Scaling to **100B parameters** would likely require 1024×A100 GPUs, which is currently beyond our capacity, but prior work (like DeepSeek-R1 with 671B parameters) suggests GRPO can effectively scale. The consistent gains from 3B to 32B models indicate our approach maintains effectiveness with increasing scale.
>
> *Table R1. Test accuracy (%) of 32B models on Erdős.*
>
> |                      |    Easy   |   Medium  |   Hard    | Challenging |  Average  |
> |----------------------|-----------|-----------|-----------|-------------|-----------|
> | Qwen2.5-32B-Instruct |   70.57   |   68.73   |   33.38   |     9.00    |   47.10   |
> |      G1-Zero-32B     | **97.79** | **93.00** | **65.25** |  **35.43**  | **75.06** |
>
>
> *Table R2. Test accuracy (%) of 32B models on math tasks.*
>
> |                      |   GSM8K   |    MATH   |
> |----------------------|-----------|-----------|
> | Qwen2.5-32B-Instruct |   90.67   |   76.80   |
> |      G1-Zero-32B     | **90.75** | **80.80** |
>
> ---
>
> **Q3.** All the experiments are run only once. The authors mentioned that randomness is negligible. LLMs are known to give different answers based on the questions. So ML practice suggests that multiple run is needed. Can you provide the robustness/correctness of the method by running the same questions multiple times on some tasks?
>
> **A3.**  To rigorously evaluate robustness, we conducted 32 repeated runs with different random seeds. The results in **Table R3** demonstrate consistently small standard deviations (<1% across all models and difficulty levels), confirming the stability of our method against potential randomness in LLM outputs. This empirical validation supports our claim about the method's reliability, and we will add the experiments in our paper.
>
> *Table R3.  Test accuracy (%) for 32 runs with different random seeds.*
> |                |   Easy    |   Medium  |    Hard   | Challenging |
> |----------------|-----------|-----------|-----------|-------------|
> | Qwen2.5-3B-Instruct |45.65 +- 0.51| 30.88 +- 0.36 | 10.36 +- 0.21 | 1.54 +- 0.29 |
> |        G1-3B        |94.96 +- 0.32| 83.22 +- 0.24 | 41.48 +- 0.40 | 7.96 +- 0.64 |
> | Qwen2.5-7B-Instruct |57.50 +- 0.13| 44.92 +- 0.03 | 19.90 +- 0.31 | 3.45 +- 0.22 |
> |        G1-7B        |95.66 +- 0.12| 88.89 +- 0.16 | 50.76 +- 0.53 | 24.46 +- 0.84|
>
> ---
>
> **Q4.** The paper does not really discuss the future directions, which is needed for the research paper. What are the future directions of this particular research?
>
> **A4.** Thanks for highlighting this important aspect. Below we provide the future directions and will add them to the paper.
>
> In future work, we aim to explore dynamic difficulty scheduling during RL training to address the sample inefficiency issue. On a broader scale, we also plan to extend our approach to the following scenarios:
> - Handling larger graphs with thousands of nodes, aligning with the long-context reasoning challenges;
> - Incorporating visual inputs (*e.g.* images depicting graphs)  to enhance real-world applicability;
> - Adapting G1 to more practical domains such as logistics, knowledge graph reasoning, and tabular problem-solving, where structured reasoning is critical.
>
>
> ---
>
> **Q5.** Limitations and societal impacts are not clear. I suggest authors make them clear. They mentioned that the limitations are in experiment section, which I don't see them. Societal impacts are supposed to be in appendix, again I don't see them. Can authors make them clearer, maybe in a separate section? They can be included with future directions that I already suggested to add.
>
> **A5.**  We appreciate the reviewer’s suggestion to clarify limitations and societal impacts. Below we provide a dedicated discussion and will add it to the paper.
>
> **Limitations:**  While G1 demonstrates significant improvements in graph reasoning through RL training, it inherits GRPO's sample inefficiency and requires extensive rollouts for challenging tasks (_e.g._, NP-hard problems), which might be solved by dynamic difficulty scheduling during training.  Although G1 demonstrates strong generalization to real-world graph tasks, _e.g._, node classification and link prediction, its generalization to highly domain-specific applications (_e.g._, molecular property prediction) and other structured data (_e.g._, tabular, time-series) remains untested.
>
> **Social Impacts:** G1’s graph reasoning capabilities offer practical benefits for education (e.g., adaptive graph theory tutoring) and scientific research. While the technology could amplify existing biases in training data, these risks can be mitigated through careful dataset design and human oversight. When developed responsibly, G1 has the potential to support human experts in solving complex graph problems.
>
> -----
> Thanks for your comments and hope our answers could address your concerns. Please let us know if there is more to clarify. We are happy to address them during the discussion stage.

---

> > ### Comment · Reviewer_2Hnx · 2025-08-05
> > **Response to Rebuttal**
> >
> > I would like to thank the authors for the extensive rebuttal with a short period of time. All of my concerns are addressed. Ib believe that the paper should be accepted and it will be very helpful for the research community. Therefore, I'm increasing my score to strong accept.

---

> > > ### Author Response · Authors · 2025-08-07
> > >
> > > Thank you very much for your insightful reviews and strong support of our work. We appreciate the time and effort you dedicated to evaluating our work and the positive feedback about our work.

---

### Official Review · Reviewer_muF5 · 2025-07-02

**Clarity:** 3
**Significance:** 4
**Originality:** 3
**Rating:** 5
**Confidence:** 4

**Summary:**

This paper presents G1, using reinforcement learning to teach LLMs to solve graph theoretic problems. First, a new dataset collection is constructed called Erdos (I love this name by the way), which provides an amount of graph-related tasks serving various reasoning tasks. The RL procedure follows commonly used GRPO. From the results G1 is very strong and could even beat larger models by reinforcing a small model.

**Questions:**

Q1. Do you think there is any unique graph-specific things that can be leverages to improve the design of GRPO training on graphs?

Q2. I love the name of the new dataset, but let's be careful. Graph theory and graph-theoretic reasoning is different. Some clarifications could be made here (not affecting the quality of the paper).

**Ethical Concerns:**

["NO or VERY MINOR ethics concerns only"]

**Final Justification:**

The authors are able to address all my concerns, and I feel confident to recommend an acceptance for this manuscript. It may not possess much real-world impact at this time (and that's why I'm not confident to recommend a strong accept), but this manuscript should possess great academic potential to inspire future works as long as everything is made open-sourced and transparent.

**Limitations:**

N/A. No remarkable negative societal impact.

**Quality:**

4

**Strengths And Weaknesses:**

Strength: Very recent R1 technique on graphs with exciting results. The presentation is clear. It is shown that G1 can even be generalized beyond graphs.

W1. Line 32 "Different from prior work, we believe LLMs pretrained on Internet-scale data already possess graph reasoning ability." Dangerous! This is a very strong claim without any supporting evidence in these two lines.

W2. The ending of introduction is inspiring yet not very clear. I would suggest more discussions on how to leverage G1 to build graph foundation models. This paper presents a grand position on graph foundation model (that there could be a paradigm shift with RL) but lacks supporting evidence and discussions.

W3. A new dataset is presented, but only reported some details and the scales. The presented dataset, however, is not evaluated. I just have no idea of what the constructed dataset looks like, and if different prompt choice would affect evaluating the LLMs. In other words, how good and robust is the constructed dataset?

W4. The proposed reinforcement learning technique is a data-centric method from my perspective. The most important thing is an amount of available graph-theoretic data with correct labels/ground-truth. The training is a mimic of GRPO, which means G1 could suffer from draw backs of GRPO, such as sampling inefficiency and optimizing instability. I don't see (or maybe I missed) study on these in 5.3 training analysis or appendix.

W5. I just cannot intuitively understand how conducting G1 on graph theoretic reasoning tasks could improve beyond graphs, such as GSM8K and MATH. Given the distribution shift in the training data and test data, I would assume the performance to degrade, which is not the case. More discusion/clarification here is needed (would be better with some discussion also on non-graph-theoretic task generalizations).

W6. Why don't you compare with GPT-4o or other SOTA LLMs? It's ok if not outperforming those models as they might have a very large number of parameters, but it would provide more evidence of how good G1-7B is achieving.

W7. Nothing on efficiency, such as training time, is reported.

---

> ### Author Rebuttal · Authors · 2025-07-31
>
> We are grateful to the reviewer for acknowledging our recent technique and its exciting results, for finding the presentation clear, and for highlighting our model's impressive generalization beyond the graph domain. Below, we present a detailed response to each of your concerns.
>
> -----
>
> **Q1.** Line 32 "Different from prior work, we believe LLMs pretrained on Internet-scale data already possess graph reasoning ability." Dangerous! This is a very strong claim without any supporting evidence in these two lines.
>
> **A1.** Thanks for bringing it to our attention! Indeed, this is a strong claim that is hard to exactly verify. We will tune it down following your advice, e.g., "we **reckon** LLMs pretrained on Internet-scale data **might** already possess graph reasoning ability."  Meanwhile, we do observe some supportive evidence for this speculation. As shown in **Table R1** below, the graph-learning reward can improve by 51% in just 50 steps (0.350→0.529), which is rarely possible if the pretrained model has not learned relevant knowledge at all. We will add this discussion in the revision.
>
> *Table R1. The tendency of rewards across the RL training steps.*
>
> |  Step  |   1   |  50   |  100  |  150  |  200  |  250  |   300  |
> |--------|-------|-------|-------|-------|-------|-------|--------|
> | Reward | 0.350 | 0.529 | 0.598 | 0.643 | 0.660 | 0.686 |  0.700 |
>
> ---
>
> **Q2.** The ending of introduction is inspiring yet not very clear. I would suggest more discussions on how to leverage G1 to build graph foundation models. This paper presents a grand position on graph foundation model (that there could be a paradigm shift with RL) but lacks supporting evidence and discussions.
>
> **A2.** Thanks for bringing it up. Through the final paragraph, we want to show that G1 itself demonstrates a new path for building graph foundation models via RL training of LLMs. More concretely, it consists of the following three key model choices:
> -   **Pretraining:**  Rather than training specific graph models (e.g., GNNs) from scratch with graph data, we can leverage LLMs pretrained with Internet-scale data and capable of general-purpose problem solving.
> -   **RL training:**  Instead of direct supervised finetuning on graph data, we use RL to incentivize reasoning processes via LLMs themselves for solving graph tasks.
> -   **Synthetic Graph Tasks:** Due to the limited amount of graph data, we can adopt diverse synthetic graph tasks (such as those in Erdős) for RL training of LLMs, which not only overcomes data scarcity but also gives rigorous learning rewards.
>
> The success of R1 demonstrates the effectiveness and scalability of this approach and thus we believe that it can be adopted for further building powerful graph foundation models. We will add this discussion to the paper.
>
> ---
>
> **Q3.** A new dataset is presented, but only reported some details and the scales. The presented dataset, however, is not evaluated. I just have no idea of what the constructed dataset looks like, and if a different prompt choice would affect evaluating the LLMs. In other words, how good and robust is the constructed dataset?
>
> **A3.** We want to note that due to the limit of space, we have included some details in the appendix.  Specifically, for  **dataset composition**, _Appendix E.2_ provides a complete task list in Erdős, with representative prompts and solutions for each task type. We hope this table can help readers to have an overall grasp of the dataset and will publicly release the datasets later. For **prompt robustness**, we rigorously test prompt sensitivity by having GPT-4o generate three semantically equivalent prompt variants. _Table R2_ shows minimal performance variance (<1.5% standard deviation) across all models and difficulty levels, confirming our benchmark’s stability to phrasing changes. We will also include these results in the paper for completeness following your advice.
>
> *Table R2. Test accuracy (%) on different prompts, mean and standard deviation reported.*
> |           ~     |   Easy    |   Medium  |    Hard   | Challenging |
> |----------------|-----------|-----------|-----------|-------------|
> | Qwen2.5-3B-Instruct |44.26 +- 0.63|30.52 +- 1.08|10.71 +- 0.41|1.24 +- 0.07|
> |        G1-3B        |94.86 +- 0.69|83.03 +- 0.42|41.08 +- 1.35|9.67 +- 1.05|
> | Qwen2.5-7B-Instruct |56.48 +- 1.17|56.48 +- 1.17|18.81 +- 0.79|3.10 +- 0.18|
> |        G1-7B        |95.48 +- 0.18|88.39 +- 0.67|51.27 +- 0.13|25.14 +- 1.04|
>
> ----
>
> **Q4.** The proposed reinforcement learning technique is a data-centric method from my perspective. The most important thing is an amount of available graph-theoretic data with correct labels/ground-truth. The training is a mimic of GRPO, which means G1 could suffer from drawbacks of GRPO, such as sampling inefficiency and optimizing instability. I don't see (or maybe I missed) study on these in 5.3 training analysis or appendix.
>
> **A4.** For **sampling inefficiency,** G1 encounters the issue on tasks with **challenging** difficulty (*e.g.,* NP-hard problem), where initially sampling a correct reasoning path is hard. Despite million-scale rollouts, G1-7B achieves less than 25% accuracy on these tasks on average. We hypothesize that adaptive difficulty scheduling during training could help mitigate this issue. For **optimizing instability**, shown in _Table R3_, G1 exhibits stable RL training. There is a consistent 51% reward improvement within 50 epochs and a monotonic progression to 0.700.  Full reward trajectory plots will be provided in the paper. We will include these discussions in our paper, and we believe further development of better RL training algorithms on resolving these issues could also benefit G1 training as well.
>
> *Table R3. The tendency of rewards across the RL training steps.*
>
> |  Step  |   1   |  50   |  100  |  150  |  200  |  250  |   300  |
> |--------|-------|-------|-------|-------|-------|-------|--------|
> | Reward | 0.350 | 0.529 | 0.598 | 0.643 | 0.660 | 0.686 |  0.700 |
>
> ----
>
> **Q5.** I just cannot intuitively understand how conducting G1 on graph theoretic reasoning tasks could improve beyond graphs, such as GSM8K and MATH. Given the distribution shift in the training data and test data, I would assume the performance to degrade, which is not the case. More discussion/clarification here is needed (would be better with some discussion also on non-graph-theoretic task generalizations).
>
> **A5.** This is a very interesting question! In fact, the transferability between different reasoning domains has also been observed in other works and been actively under research. For example, [1] demonstrates that reinforcement learning exhibits strong generalization even under distribution shifts, as it encourages models to learn robust, task-agnostic reasoning strategies rather than overfitting to surface-level patterns. Additionally, RL training has been shown to foster mutual reinforcement across seemingly distinct tasks, *e.g.*, optimizing on puzzles can help reasoning on math [2]. Similarly, due to the similarity between math and graph problems, they would also require similar reasoning skills (such as guessing, backtracking, verification). Therefore, solving graph problems could have positive transferability to math domains.
>
> [1]. Chu, Tianzhe, et al. "Sft memorizes, rl generalizes: A comparative study of foundation model post-training." arXiv preprint arXiv:2501.17161 (2025).
>
> [2]. Li, Yu, et al. "Can One Domain Help Others? A Data-Centric Study on Multi-Domain Reasoning via Reinforcement Learning" arXiv preprint arXiv:2507.17512 (2025).
>
> ---
>
> **Q6.** Why don't you compare with GPT-4o or other SOTA LLMs? It's ok if not outperforming those models as they might have a very large number of parameters, but it would provide more evidence of how good G1-7B is achieving.
>
> **A6.** We did not include GPT-4o as comparison for its high cost on API calling. As an alternative,  we include **OpenAI o3-mini**, that, according to [official OpenAI reports](https://openai.com/zh-Hans-CN/index/openai-o3-mini/), significantly outperforms GPT-4o on **reasoning benchmarks**. Our comparison with OpenAI o3-mini on graph reasoning tasks (_Table 2_) demonstrates that G1-7B achieves competitive performance despite its smaller scale.
>
> ---
>
> **Q7.** Nothing on efficiency, such as training time, is reported.
>
> **A7.** Thank you for raising this point. G1-7B was trained in approximately **40 hours on 8×A100 GPUs**, while G1-3B required around **35 hours on 4×A100 GPUs**. We will include these details in the paper.
>
> ---
>
> **Q8.** Do you think there is any unique graph-specific things that can be leveraged to improve the design of GRPO training on graphs?
>
> **A8.** While graph structural priors could be explicitly incorporated into the LLM architecture, this might compromise its generality. Instead, we implicitly encode these priors through **reward design**. Specifically, graph-theoretic tasks offer unique advantages for GRPO training, particularly in  **controllable difficulty scaling**. Unlike traditional math reasoning benchmarks—where difficulty is often subjective and requires complex data curation—graph problems allow precise difficulty adjustment in two key ways: (1) theoretical complexity: we can select problems across complexity classes (e.g., linear-time vs. NP-hard) to create a structured curriculum, and (2) graph size: even within the same task, difficulty can be tuned by varying the number of nodes/edges. This flexibility enables more dynamic training strategies, such as  **reward reweighting**, which we have studied in _Appendix D_, and  **adaptive difficulty scheduling**, which can be explored in future study.

---

> > ### Comment · Reviewer_muF5 · 2025-08-01
> > **Please stay tuned!**
> >
> > Dear authors,
> >
> > Thank you for your detailed rebuttal. I will be traveling over the next couple of days and may not be able to check the comments during that time. However, I will make sure to carefully read your response, along with the other reviewers' comments, and provide my additional feedback ideally before August 4th.
> >
> > Best regards,
> > Anonymous reviewer

---

> > ### Comment · Reviewer_muF5 · 2025-08-06
> >
> > Dear authors,
> >
> > Thank you for your patience I finally get some time to look over the rebuttal carefully.
> >
> > W1. Maybe you can try the same algorithm on, randomly initialized transformer, which does not contain any knowledge, i suppose. Let me know if it is not applicable.
> >
> > W2. Clear. Thanks!
> >
> > W3. Clear. i do see the other reviewer mentioning about the number of the nodes in the dataset is 35 at max. You have your response by generating graphs with more nodes. Just want to confirm your pipeline should be able to generate any size? And  a follow-up: what might be the problem for generating many large graphs (i.e., why didn't you do graphs with large number of nodes in the current manuscript)?
> >
> > W4. Clear. Thanks!
> >
> > W5. I would say mostly clear. I need to check those two papers. The explanation generally makes sense to me.
> >
> > W6. I understand the cost to conduct large-scale experiments, but maybe a little bit comparison on partial of the data (which will cost less) just to know how good your models are. I know o3-mini has better reasoning capability but people are mostly using 4o.
> >
> > W7. Clear. Thanks!.
> >
> > W8. Clear. Thanks!
> >
> > I'm more confident that this paper should lean towards acceptance (raising my confidence to 5). For Reviewer R7SY's opinion on minimal real-world impact, I would say, the R1-style research on graphs are not explored yet, and from my research taste, we should not request deployment and real-world impact at this point. I'm confident this paper would have academic impact as long as everything is open-sourced. However, this also prevents me of raising the score to, a strong acceptance.

---

> > > ### Author Response · Authors · 2025-08-07
> > >
> > > We thank Reviewer muF5 for the active engagement in the discussion and the thoughtful response to our rebuttal. Below, we address the further concerns.
> > >
> > > -----
> > >
> > > **Q1.** Maybe you can try the same algorithm on, randomly initialized transformer, which does not contain any knowledge, i suppose. Let me know if it is not applicable.
> > >
> > > **A1.** Thank you for your helpful suggestion. We are currently conducting the experiments and will report the results here once they are available.
> > >
> > > -----
> > >
> > > **Q2.** I do see the other reviewer mentioning about the number of the nodes in the dataset is 35 at max. You have your response by generating graphs with more nodes. Just want to confirm your pipeline should be able to generate any size? And a follow-up: what might be the problem for generating many large graphs (i.e., why didn't you do graphs with large number of nodes in the current manuscript)?
> > >
> > > **A2.** (1) Through experiments on graphs with 35–100 nodes, we demonstrate that our  G1  models can generalize to  _unseen_  graph sizes and higher difficulties (in general, larger graphs require longer reasoning paths).
> > >
> > > (2) Currently, we limit our analysis to smaller graph because of **the context window limit of underlying LLMs** (_e.g._, Qwen2.5-7B-Instruct) . The token count scales _quadratically_ with the number of nodes. As shown in **Table R1**, a ~200-node graph often exceeds 32k tokens, surpassing the maximum  _effective_ context window of many open-source LLMs [1]. Long-context understanding is actively studied in the LLM literature and there are some cutting-edge proprietary variants (_e.g._, OpenAI’s GPT-4.1) supporting inputs of over 1M tokens (though graphs of  2000 nodes need ~4M). Due to computational constraints (demanding a **huge** GPU memory), it is hard for us to evaluate on very long context. In this paper, we establish the roadmap of G1 and validate its effectiveness, we believe our approach can be scaled to larger graphs with the rapid progress of long-context studies.
> > >
> > > *Table R1. Token numbers of graph with different node sizes. We generate random graphs by Erdős–Rényi model with an edge probability of 0.2. For each node number, we generate 10 graphs and reports the mean and standard derivation of the token numbers. For tokenization, we utilize the tokenizer of Qwen2.5-Instruct.*
> > >
> > > | Node Number | 30 | 50 | 100 | 200 | 500 | 1000 | 2000 |
> > > |----|-----|------|--------|-------|-------|------|-----|
> > > | Token Number| 641.4 ± 58.3 | 1941.5 ± 51.6 | 7842.8 ± 285.5 | 35694.1 ± 574.2 (>32k) | 237391.4 ± 895.3 (>200k) |  977858.9 ± 2329.9 (~1M)| 4355340.7 ± 3997.0 (~4M)|
> > >
> > > [1]. Hsieh, Cheng-Ping, et al. "RULER: What's the Real Context Size of Your Long-Context Language Models?." COLM, 2024.
> > >
> > >
> > > **Q3.** I understand the cost to conduct large-scale experiments, but maybe a little bit comparison on partial of the data (which will cost less) just to know how good your models are. I know o3-mini has better reasoning capability but people are mostly using 4o.
> > >
> > > **A3.** We randomly sample 20 examples per task from Erdős’s evaluation set to construct a subset of 1,000 samples. As shown in **Table R2**, G1-3B performs comparably to GPT-4o on average (57.37% vs. 55.13%), while G1-7B surpasses GPT-4o across all difficulty levels, **exceeding it by over 10% on average**. This comparison further validates the strong graph reasoning capabilities of the G1 models. We appreciate the constructive suggestion and will include these results in the revision.
> > >
> > > *Table R2. Test accuracy (%) of GPT-4o and G1 variants on a subset of Erdős.*
> > > |   ~  |   Easy    |   Medium  |    Hard   | Challenging | Average |
> > > |--------|--------|-----------|-----------|----------|-------|
> > > | GPT-4o-2024-11-20 | 82.50 | 81.82 | 44.06 | 12.14 | 55.13 |
> > > |  G1-3B  | **96.43** | 85.45 | 41.88 | 5.71  | 57.37 |
> > > |  G1-7B  | **96.43** | **88.64** | **52.50** | **23.57** | **65.29** |
> > >
> > > -------

---

> > > ### Author Response · Authors · 2025-08-07
> > >
> > > **Q4.** For Reviewer R7SY's opinion on minimal real-world impact, I would say, the R1-style research on graphs are not explored yet, and from my research taste, we should not request deployment and real-world impact at this point. I'm confident this paper would have academic impact as long as everything is open-sourced. However, this also prevents me of raising the score to, a strong acceptance.
> > >
> > > **A4.** We sincerely appreciate your recognition of the value and potential of our work. Beyond the node classification and link prediction validated in the paper, we present a specific fraud detection case below, in which G1 shows the potential for solving other real-world scenarios. We hope this work serves as an inspiring foundation for future advancements in specialized domains, _e.g._, knowledge graphs, protein interactions, and tabular data analysis, and we will actively open-source our codes, benchmarks, and evaluation logs to the community.
> > >
> > >
> > > **A Case for Fraud Detection**.
> > >
> > > > **Query.** You are given a fraud detection scenario modeled as a graph, where nodes represent entities (users, cards, transactions, merchants) and edges represent relationships between them. Your goal is to analyze the graph structure and identify fraudulent transactions based on patterns (e.g., unusual locations, rapid spending, or abnormal merchant behavior).
> > > > - **Nodes & Attributes**
> > > > 1. Users: - `U1`: {Name: Alice, HomeCity: CityA, UsualSpend: $100–$500} - `U2`: {Name: Bob, HomeCity: CityB, UsualSpend: $50–$300}
> > > > 2. Credit Cards: - `C1`: {User: U1, Bank: BankA, DailyLimit: $5,000} - `C2`: {User: U2, Bank: BankB, DailyLimit: $3,000}
> > > > 3. Transactions: - `T1`: {Card: C1, Amount: $200, Time: 10:00, Merchant: M1, Location: CityA} - `T2`: {Card: C1, Amount: $1,800, Time: 10:05, Merchant: M2, Location: CityC} - `T3`: {Card: C2, Amount: $2,500, Time: 11:00, Merchant: M2, Location: CityC}
> > > > 4. Merchants: - `M1`: {Name: "SafeShop", Category: Grocery, RiskScore: Low} - `M2`: {Name: "LuxuryDeals", Category: Jewelry, RiskScore: High}
> > > >-  **Edges (Relationships)**
> > > > 1. Ownership: - `U1 → OWNS → C1` - `U2 → OWNS → C2`
> > > > 2. Transactions: - `C1 → USED_IN → T1`, `C1 → USED_IN → T2` - `C2 → USED_IN → T3`
> > > > 3. Merchant Links: - `T1 → AT → M1`, `T2 → AT → M2`, `T3 → AT → M2`
> > > >
> > > > **Ground Truth.** T2 and T3.
> > > >
> > > > **Response of G1-7B.**
> > > > To identify potential fraudulent transactions in the given graph, we need to analyze the patterns and relationships between the nodes to detect any unusual behavior. Here are the details of the nodes and their relationships: (omitted for clarity)
> > > >  Let's analyze each transaction to identify any potential fraud:
> > > >  1. **Transaction T1:**  - Card: C1 - Amount: $200  - Time: 10:00 - Merchant: M1 - Location: CityA  - User: U1 - Merchant M1 is a grocery store with low risk. - This transaction is within U1's usual spending range and in their home city. It does not appear to be fraudulent.
> > > >  2. **Transaction T2:** - Card: C1 - Amount: $1,800 - Time: 10:05 - Merchant: M2 - Location: CityC  - User: U1 - Merchant M2 is a jewelry store with high risk. - This transaction is significantly larger than U1's usual spending range and occurs in a different city. It is also from a high-risk merchant. This is a potential fraud.
> > > >  3. **Transaction T3:** - Card: C2 - Amount: $2,500 - Time: 11:00 - Merchant: M2 - Location: CityC - User: U2 - Merchant M2 is a jewelry store with high risk. - This transaction is also significantly larger than U2's usual spending range and occurs in a different city. It is also from a high-risk merchant. This is a potential fraud.
> > > >
> > > >  Based on the analysis, the following transactions are identified as potential fraudulent:
> > > >  - **T2**: $1,800 spent at M2 in CityC by U1.
> > > > - **T3**: $2,500 spent at M2 in CityC by U2.
> > > >
> > > > These transactions are flagged because they involve large amounts, occur in different locations from the users' usual spending patterns, and involve high-risk merchants.
> > >
> > >
> > > ------
> > >
> > > Please let us know if there are any additional aspects that require clarification or further discussion.

---

> > > > ### Comment · Reviewer_muF5 · 2025-08-07
> > > >
> > > > Q1: Yes. Please let me know the results, if applicable before the end of rebuttal.
> > > >
> > > > Q2: I'm clear. Sad to know context window is the limitation, but maybe that's also some future direction for researchers to work on (how to make g1 generalize to LLMs with limited context window).
> > > >
> > > > Q3: That's great to know. Please include those results in the paper.
> > > >
> > > > Q4: I wasn't expecting new results on this but just sharing my option. But thanks for the new experimental results.
> > > >
> > > > I'm more confident now to increase my rating and vote for an acceptance (with score 5 and confidence 4). I appreciate the author's effort for engaging in the rebuttal and providing new results that can address my concerns. Still, everything is more like prototypes and may not have that much real-world impact by this time, so I don't feel confident to recommend a strong accept.

---

> > > > > ### Author Response · Authors · 2025-08-08
> > > > >
> > > > > Thanks for your patient and prompt reply.
> > > > >
> > > > > For **Q1**, we perturb the LLM backbone's weights by adding random Gaussian noise, which is expected to disrupt the model's pretrained knowledge. Since the original weights' standard deviation ranges between 0.01 and 0.04, we set the noise's standard deviation to 0.01 to avoid excessive distortion. So far, the RL training has progressed for approximately 50 steps. However, the mean rewards have **remained at zero since the beginning**. Upon inspection, we find that the model initially fails to generate **any** correct answers during rollouts, resulting in no meaningful rewards for RL exploration (which might also occur with a randomly initialized transformer). From these results, we cannot assert whether LLMs pretrained on Internet-scale data inherently possess graph reasoning ability. Investigating this further—for instance, by designing counterfactual graph tasks—could provide clearer insights, though such an analysis would exceed our current time constraints.
> > > > >
> > > > > We thank you for the suggestion again and will moderate the claim in Line 32 until we gather more robust evidence to support it.

---

### Official Review · Reviewer_R7SY · 2025-07-02

**Clarity:** 3
**Significance:** 2
**Originality:** 3
**Rating:** 4
**Confidence:** 4

**Summary:**

This paper introduces a novel dataset covering more than 50 graph-theoretic tasks with varying levels of difficulty. Leveraging this dataset, they present G1, the first large language model trained with reinforcement learning for graph computational problems. G1 demonstrates superior performance compared to existing graph-focused language models on graph-related tasks, and exhibits zero-shot generalization abilities to non-graph domains such as mathematics.

**Questions:**

1. In the reward design, the authors mainly discuss outcome-based rewards. Have the authors considered designing format rewards or intermediate-step rewards, or assigning different weights to various types of rewards? Would such modifications lead to improved performance?
2. Could the authors provide a plot showing the relationship between RL training steps and response length? This would offer insights into the learning dynamics and solution complexity.
3. In the generalization experiments (e.g., mathematics), does G1 exhibit longer chain-of-thought (CoT) responses or changes in reasoning patterns, such as increased stepwise thinking? Some analysis or examples would be helpful to understand the model's behavior in out-of-domain tasks.

**Ethical Concerns:**

["NO or VERY MINOR ethics concerns only"]

**Final Justification:**

The further response has addressed my concerns in some extent. I am neutral to this work.

**Limitations:**

Yes.

**Paper Formatting Concerns:**

No.

**Quality:**

2

**Strengths And Weaknesses:**

**Strengths:**

1. This paper is the first to employ reinforcement learning (RL) to train large language models (LLMs) for solving graph computational problems, and demonstrates performance that surpasses previous graph-focused LLMs.
2. The authors propose a dataset that covers a wide range of graph-theoretic tasks with clearly defined difficulty levels, offering a valuable resource for both training and evaluation.
3. The presented G1 model exhibits impressive zero-shot generalization capabilities, not only across unseen graph tasks but also in non-graph domains such as mathematics.

**Weaknesses:**

1. The experiments are limited to graphs with a maximum of 35 nodes, which is relatively small. It would be more convincing to see results on graphs with at least 60 nodes to better assess scalability.
2. The paper lacks any analysis or statistics regarding the number of output tokens generated by the model, which could be important for understanding efficiency and solution complexity.
3. The technical novelty appears somewhat limited. The reward design focuses solely on outcome-based rewards, which is relatively naïve; more sophisticated reward structures could be explored.

---

> ### Author Rebuttal · Authors · 2025-07-31
>
> We thank Reviewer R7SY for recognizing our work as a pioneering effort that surpasses previous models, for acknowledging the value of our comprehensive dataset, and for highlighting our model's impressive zero-shot generalization capabilities. Below is our response to your comments.
>
> -------
>
> **Q1.** The experiments are limited to graphs with a maximum of 35 nodes, which is relatively small. It would be more convincing to see results on graphs with at least 60 nodes to better assess scalability.
>
> **A1.** That is a very good point! Follow your suggestion, we construct a new test set of 5,000 graphs with 36-100 nodes, with other settings kept the same, which ensures there is no overlap between training and test data. **Table R1** below shows that both G1-3B and G1-7B achieve strong zero-shot generalization to these larger graphs **without additional training**, significantly outperforming the baselines across difficulty levels. These results demonstrate our method's effective scalability beyond the original training distribution.
>
> *Table R1. Zero-shot generalization (accuracy in percentage) of G1 to larger graphs with **36-100 nodes**.*
> |        ~        |   Easy    |   Medium  |    Hard   | Challenging |  Average  |
> |----------------|-----------|-----------|-----------|-------------|-----------|
> | Qwen2.5-3B-Instruct |  27.98   |   28.53   |    5.26    |    0.29   |   16.74   |
> |        G1-3B        | **79.39** | **65.66** | **18.46** | **3.74**  | **44.29** |
> | Qwen2.5-7B-Instruct |   37.86   |   41.56   |    9.17   |   1.17    |   23.94   |
> |        G1-7B        | **76.65** | **70.67** | **23.16** |  **5.22** | **46.46** |
>
> ---
>
> **Q2.** The paper lacks any analysis or statistics regarding the number of output tokens generated by the model, which could be important for understanding efficiency and solution complexity. Could the authors provide a plot showing the relationship between RL training steps and response length? This would offer insights into the learning dynamics and solution complexity.
>
> **A2.** Thanks for your suggestion. While we cannot add a plot per the NeurIPS rebuttal constraint, we provide the average response lengths across different steps in **Table R2**. The data shows relatively stable response lengths throughout training (813-962 tokens), suggesting consistent solution complexity and stable generation behavior. We'll include a complete plot in the paper.
>
> *Table R2. Average response length across the RL training steps on Erdős.*
> |  Step |   1   |  50  |  100   |   150   |  200   |   250   |   300   |
> |--------|-------|--------|--------|--------|--------|-------|--------|
> | Average response Length  | 813 | 869 | 883 | 854 | 889 | 962 | 933 | 956 |
>
> ---
>
> **Q3.** The technical novelty appears somewhat limited. The reward design focuses solely on outcome-based rewards, which is relatively naïve; more sophisticated reward structures could be explored. In the reward design, the authors mainly discuss outcome-based rewards. Have the authors considered designing format rewards or intermediate-step rewards, or assigning different weights to various types of rewards? Would such modifications lead to improved performance?
>
> **A3.** We understand the reviewer’s concern regarding the absence of sophisticated rewards (_e.g._, intermediate-step rewards). However, to integrate graph reasoning into the current RL framework, we carefully designed **graph-specific rewards** that go beyond the original GRPO’s scope—which was tailored for math tasks with unique solutions.
>
> Our paper introduces graph task-specific rewards with three key innovations:
> - For tasks with unordered-set answers (_Section 4.1_), we replace strict value matching with **flexible set matching**, granting intermediate rewards for partially correct solutions.
> - For non-unique solutions, we implement **task-specific verification algorithms** to rigorously assess correctness.
> - As detailed in  _Appendix D_, we apply a  _soft_  data mixture strategy dynamically **reweights rewards by task difficulty**, yielding better average performance than G1-3B.
>
> Additionally, we adopt PRIME [1]—an intermediate-reward-based RL method that calculates token-level dense rewards to estimate advantages. As shown in **Table R3**, G1-PRIME-3B exhibits consistent improvement over the Qwen2.5 baseline, however, it still lags slightly behind the G1-3B (55.94% vs. 59.76% on average), indicating the effectiveness of our paper’s reward design.
>
> *Table R3. Test performance (%)  on Erdős benchmark.*
> |     ~    |   Easy   |   Medium  |   Hard   | Challenging |  Average  |
> |----------|----------|-----------|----------|-------------|-----------|
> | Qwen2.5-3B-Instruct |  45.71  |  30.18  |   9.44  |  1.29  |  22.72  |
> |      G1-PRIME-3B    |  88.79  |  78.73  |  38.12  |  8.57  |  55.94  |
> |        G1-3B        |**94.86**|**84.64**|**41.25**|**7.57**|**59.76**|
>
> [1]. Cui, Ganqu, et al. "Process reinforcement through implicit rewards." arXiv preprint arXiv:2502.01456 (2025).
>
> ---
>
> **Q4.** In the generalization experiments (e.g., mathematics), does G1 exhibit longer chain-of-thought (CoT) responses or changes in reasoning patterns, such as increased stepwise thinking? Some analysis or examples would be helpful to understand the model's behavior in out-of-domain tasks.
>
> **A4.** Thanks for your insightful question! Following your suggesstion, we calculate the average response length on the MATH dataset. As shown in **Table R4**, G1 does not exhibit significant length variation compared to baselines, suggesting that its generalization to non-graph tasks (e.g., GSM8K and MATH) stems not from longer chains of thought (CoT). We reckon the improvement, e.g., in the G1-7B case, is caused by the inherent similarity between math problems and graph problems --in broad terms, graph-theoretic problems are math problems as well. Therefore, learning to solve graph problems can teach the models effective techniques to solve math problems as well.
>
>
> *Table R4. Average response length on MATH dataset.*
> | Model | Average response length |
> |--------|------------------------|
> | Qwen-3B-Instruct | 612 |
> |  G1-3B | 663 |
> | Qwen-7B-Instruct | 646 |
> | G1-7B  | 586 |
>
> -----
>
> We appreciate your time in reviewing our work and hope these responses resolve your queries. Please let us know if there are any other points you wish to discuss.

---

> > ### Comment · Reviewer_R7SY · 2025-08-05
> > **Thank you for your detailed rebuttal.**
> >
> > After careful review, we have decided to **maintain our original score** for the following reasons:
> >
> > 1. **Superficial Generalization Claims**: While the reported improvements on math benchmarks are noted, the manuscript fails to provide any substantive analysis of the underlying mechanisms. Simply attributing gains to the assertion that "graph problems are essentially math problems" is unconvincing and does not constitute a meaningful explanation. There is a lack of evidence that the model has developed any deeper or more transferable reasoning abilities, particularly as no analysis of reasoning patterns or stepwise thinking is presented.
> >
> > 2. **Minimal Real-World Value**: The model's applicability is severely limited by its inability to handle graphs larger than approximately 100 nodes—a scale far below that required in most practical or industrial contexts. Moreover, the generalization demonstrated outside the graph domain is limited, and thus the work offers little benefit for real-world deployment.
> >
> > In conclusion, despite technical soundness and some interesting results, the work lacks depth in both the analysis of generalization and in practical relevance. These substantial shortcomings prevent me from raising the score.

---

> > > ### Author Response · Authors · 2025-08-07
> > >
> > > **Q2.**  **Minimal Real-World Value**: The model's applicability is severely limited by its inability to handle graphs larger than approximately 100 nodes—a scale far below that required in most practical or industrial contexts. Moreover, the generalization demonstrated outside the graph domain is limited, and thus the work offers little benefit for real-world deployment.
> > >
> > > **A2.** (1). Currently, we limit our analysis to smaller graph because of **the context window limit of underlying LLMs** (_e.g._, Qwen2.5-7B-Instruct) . The token count scales _quadratically_ with the number of nodes. As shown in **Table R1**, a ~200-node graph often exceeds 32k tokens, surpassing the maximum  _effective_ context window of many open-source LLMs [1]. Long-context understanding is actively studied in the LLM literature and there are some cutting-edge proprietary variants (_e.g._, OpenAI’s GPT-4.1) supporting inputs of over 1M tokens (though graphs of  2000 nodes need ~4M). Due to computational constraints (demanding a **huge** GPU memory), it is hard for us to evaluate on very long context. In this paper, we establish the roadmap of G1 and validate its effectiveness, we believe our approach can be scaled to larger graphs with the rapid progress of long-context studies.
> > >
> > > *Table R1. Token numbers of graph with different node sizes. We generate random graphs by Erdős–Rényi model with an edge probability of 0.2. For each node number, we generate 10 graphs and reports the mean and standard derivation of the token numbers. For tokenization, we utilize the tokenizer of Qwen2.5-Instruct.*
> > >
> > > | Node Number | 30 | 50 | 100 | 200 | 500 | 1000 | 2000 |
> > > |----|-----|------|--------|-------|-------|------|-----|
> > > | Token Number| 641.4 ± 58.3 | 1941.5 ± 51.6 | 7842.8 ± 285.5 | 35694.1 ± 574.2 (~35k) | 237391.4 ± 895.3 (~200k) |  977858.9 ± 2329.9 (~1M)| 4355340.7 ± 3997.0 (~4M)|
> > >
> > > [1]. Hsieh, Cheng-Ping, et al. "RULER: What's the Real Context Size of Your Long-Context Language Models?." COLM, 2024.

---

> ### Author Response · Authors · 2025-08-05
>
> Dear Reviewer,
>
> Thank you once again for your thoughtful reviews and for taking the time to engage with our work. As the author-reviewer discussion period is nearing its close, we wanted to check in and see if you have any further questions or require additional clarifications based on our rebuttal. We’re more than happy to provide any extra details that might be helpful.
>
> We sincerely appreciate your time and expertise.
>
> Best regards,
>
> The Authors

---

> ### Author Response · Authors · 2025-08-07
>
> We thank Review R7SY for the detailed reply, and we address your further concerns below.
>
> ----
>
> **Q1.**  **Superficial Generalization Claims**: While the reported improvements on math benchmarks are noted, the manuscript fails to provide any substantive analysis of the underlying mechanisms. Simply attributing gains to the assertion that "graph problems are essentially math problems" is unconvincing and does not constitute a meaningful explanation. There is a lack of evidence that the model has developed any deeper or more transferable reasoning abilities, particularly as no analysis of reasoning patterns or stepwise thinking is presented.
>
> **A1.** We further compare the generation results of G1-7B and Qwen2.5-7B-Instruct on math benchmarks case by case. As a summarization, G1-7B improves over Qwen2.5-7B-Instruct on
> - **more accurate numerical calculation** (_Instance 1_, in the next block)
> - **more comprehensive utilization of given information** (_Instance 2_, in the next block).
>
> It's reasonable since **the training of G1 includes complex numerical calculations** (_e.g._, shortest path) **and careful processing of questions** (_e.g._, traversing the whole graph descriptions to find key edges). We will release the complete response logs of G1 and other baselines on math benchmarks publicly later.
>
> We emphasize that our primary goal is to demonstrate **generalization across graph tasks**. To this end, we conduct a comprehensive study on graph domain transferability in Sections 5.2.1 and 5.2.2. The modest improvement on math tasks serves as a secondary outcome, indicating that RL training preserves the model's general reasoning ability. Thanks for the critique, and we will explore a more thorough analysis of transferability across broader domains (_e.g._, mathematics and graphs) in the future.

---

> ### Author Response · Authors · 2025-08-07
>
> We show two specific instances of GSM8k below. Main differences of G1-7B and Qwen2.5-7B-Instruct are highlighted by **Bold**.
>
> **Instance 1 (Calculation Correctness)**
> > **Question**: Toula went to the bakery and bought various types of pastries. She bought 3 dozen donuts which cost $68 per dozen, 2 dozen mini cupcakes which cost $80 per dozen, and 6 dozen mini cheesecakes for $55 per dozen. How much was the total cost?
> > **Ground Truth:** 694
> > **Answer of Qwen2.5-7B**-step-by-step solution:
> > - **Calculate the cost of the donuts:** - Toula bought 3 dozen donuts. - The cost per dozen donuts is $68. - Therefore, the total cost for the donuts is:**\[3 \text{dozen} \times 68 \text{ dollars/dozen} = 180 \text{ dollars}\]**
> > .......
> > - **Calculate the total cost:** - Add the costs of the donuts, mini cupcakes, and mini cheesecakes: \[180 \text{ dollars} + 160 \text{ dollars} + 330 \text{ dollars} = **670** \text{ dollars}\]
> >
> > **Answer of G1-7B** :  To calculate the total cost of the pastries that Toula bought, we first need to calculate the cost of each type of pastry separately and then sum them up.
> > - **Cost of Donuts:** - Toula bought 3 dozen donuts. - The cost per dozen donuts is $68. - Therefore, the total cost for the donuts is: **\[3 \text{ dozen} \times \68 \text{ per dozen} = 3 \times 68 = \$204\]**
> > -  Now, we add up the costs of all the pastries to find the total cost: \[\$204 \text{ (donuts)} + \$160 \text{ (mini cupcakes)} + \$330 \text{ (mini cheesecakes)} = \$**694**\]
>
> **Instance 2 (Utilization of Known Conditions)**
> > **Question**: On Mondays, Wednesdays, and Fridays, college student Kimo has three 1-hour  classes each day.  On Tuesdays and Thursdays, he has two 2-hour classes each day.  In one semester, there are 16 weeks of school.  In one semester, how many hours does Kimo spend attending classes?
> > **Ground Truth**: 272
> > **Answer of Qwen2.5-7B**:  Let's break down the problem step by step to find the total number of hours Kimo spends attending classes in one semester.
> > - **Calculate the number of hours Kimo spends in classes each week:** - On Mondays, Wednesdays, and Fridays, Kimo has 3 classes, each lasting 1 hour. \[3 \text{ classes} \times 1 \text{ hour/class} = 3 \text{ hours}\]- On Tuesdays and Thursdays, Kimo has 2 classes, each lasting 2 hours.\[2 \text{ classes} \times 2 \text{ hours/class} = 4 \text{ hours}\] - Therefore, the total number of hours Kimo spends in classes each week is: \[3 \text{ hours} + 4 \text{ hours} = **7** \text{ hours}\]
> > - **Calculate the number of hours Kimo spends in classes in one semester:** - There are 16 weeks in the semester. - The total number of hours Kimo spends in classes in one semester is: \[7 \text{ hours/week} \times 16 \text{ weeks} = **112**\text{ hours} \]
> >
> > **Answer of G1-7B**: To determine the total number of hours Kimo spends attending classes in one semester, we first need to calculate the number of hours he spends in classes each week and then multiply by the number of weeks in the semester.
> > - Step 1: Calculate Weekly Class Hours.
> >   - Mondays, Wednesdays, and Fridays: - Each day, Kimo has 3 classes, each lasting 1 hour. - Therefore, for each of these days, the total class time is: \[3 \text{ classes/day} \times 1 \text{ hour/class} = 3 \text{ hours/day}\] - **Since there are 3 days (Monday, Wednesday, Friday)**: \[3 \text{ hours/day} \times 3 \text{ days} = 9 \text{ hours}\]
> >   - Tuesdays and Thursdays: - Each day, Kimo has 2 classes, each lasting 2 hours. - Therefore, for each of these days, the total class time is: \[ 2 \text{ classes/day} \times 2 \text{ hours/class} = 4 \text{ hours/day}\] - **Since there are 2 days (Tuesday, Thursday)**: \[4 \text{ hours/day} \times 2 \text{ days} = 8 \text{ hours}\]
> >   - **Total Weekly Class Hours:**- Adding the hours from all the days:\[ 9 \text{ hours} + 8 \text{ hours} = **17** \text{ hours/week}\]
> > - Step 2: Calculate Total Class Hours in One Semester. - There are 16 weeks in the semester. - Therefore, the total class hours in one semester is: \[ 17 \text{ hours/week} \times 16 \text{ weeks} = **272** \text{ hours} \]

---

> ### Author Response · Authors · 2025-08-07
>
> (2). Regarding real-world impact, our work focuses on advancing the  **graph reasoning**  capabilities of LLMs. Notably, graph-related tasks have a broad importance in practical applications. For example, we demonstrate that G1 effectively addresses widely studied problems such as **node classification** and **link prediction** (Table 5), with direct relevance to real-world scenarios like fraud detection and recommendation systems. Below, we present **a specific fraud detection case** in which G1 analyzed the graph structure and accurately identified fraudulent transactions based on pattern recognition, showing the potential of G1 in solving real-world problems. As a pioneering effort in **RL-centric graph modeling**, we provide  **a feasible recipe** that has potential for broader applications—including knowledge graphs, protein interactions, and tabular data analysis. Indeed, as recognized by _Reviewer muF5_,
> > For Reviewer R7SY's opinion on minimal real-world impact, I would say, the R1-style research on graphs are not explored yet, and from my research taste, we should not request deployment and real-world impact at this point. I'm confident this paper would have academic impact as long as everything is open-sourced.
>
> We believe that this work could bring a new methodology for building graph foundation models and initiate new interests to further scale up this approach within the generic R1-style learning framework.
>
> **A Case for Fraud Detection**.
>
> > **Query.** You are given a fraud detection scenario modeled as a graph, where nodes represent entities (users, cards, transactions, merchants) and edges represent relationships between them. Your goal is to analyze the graph structure and identify fraudulent transactions based on patterns (e.g., unusual locations, rapid spending, or abnormal merchant behavior).
> > - **Nodes & Attributes**
> > 1. Users: - `U1`: {Name: Alice, HomeCity: CityA, UsualSpend: $100–$500} - `U2`: {Name: Bob, HomeCity: CityB, UsualSpend: $50–$300}
> > 2. Credit Cards: - `C1`: {User: U1, Bank: BankA, DailyLimit: $5,000} - `C2`: {User: U2, Bank: BankB, DailyLimit: $3,000}
> > 3. Transactions: - `T1`: {Card: C1, Amount: $200, Time: 10:00, Merchant: M1, Location: CityA} - `T2`: {Card: C1, Amount: $1,800, Time: 10:05, Merchant: M2, Location: CityC} - `T3`: {Card: C2, Amount: $2,500, Time: 11:00, Merchant: M2, Location: CityC}
> > 4. Merchants: - `M1`: {Name: "SafeShop", Category: Grocery, RiskScore: Low} - `M2`: {Name: "LuxuryDeals", Category: Jewelry, RiskScore: High}
> >-  **Edges (Relationships)**
> > 1. Ownership: - `U1 → OWNS → C1` - `U2 → OWNS → C2`
> > 2. Transactions: - `C1 → USED_IN → T1`, `C1 → USED_IN → T2` - `C2 → USED_IN → T3`
> > 3. Merchant Links: - `T1 → AT → M1`, `T2 → AT → M2`, `T3 → AT → M2`
> >
> > **Ground Truth.** T2 and T3.
> >
> > **Response of G1-7B.**
> > To identify potential fraudulent transactions in the given graph, we need to analyze the patterns and relationships between the nodes to detect any unusual behavior. Here are the details of the nodes and their relationships: (omitted for clarity)
> >  Let's analyze each transaction to identify any potential fraud:
> >  1. **Transaction T1:**  - Card: C1 - Amount: $200  - Time: 10:00 - Merchant: M1 - Location: CityA  - User: U1 - Merchant M1 is a grocery store with low risk. - This transaction is within U1's usual spending range and in their home city. It does not appear to be fraudulent.
> >  2. **Transaction T2:** - Card: C1 - Amount: $1,800 - Time: 10:05 - Merchant: M2 - Location: CityC  - User: U1 - Merchant M2 is a jewelry store with high risk. - This transaction is significantly larger than U1's usual spending range and occurs in a different city. It is also from a high-risk merchant. This is a potential fraud.
> >  3. **Transaction T3:** - Card: C2 - Amount: $2,500 - Time: 11:00 - Merchant: M2 - Location: CityC - User: U2 - Merchant M2 is a jewelry store with high risk. - This transaction is also significantly larger than U2's usual spending range and occurs in a different city. It is also from a high-risk merchant. This is a potential fraud.
> >
> >  Based on the analysis, the following transactions are identified as potential fraudulent:
> >  - **T2**: $1,800 spent at M2 in CityC by U1.
> > - **T3**: $2,500 spent at M2 in CityC by U2.
> >
> > These transactions are flagged because they involve large amounts, occur in different locations from the users' usual spending patterns, and involve high-risk merchants.
>
> -----
> We are available to clarify any remaining concerns or provide supplementary information as needed.

---

### Official Review · Reviewer_WDtd · 2025-07-03

**Clarity:** 3
**Significance:** 2
**Originality:** 2
**Rating:** 4
**Confidence:** 2

**Summary:**

This work first compiles a large-scale graph-theoretic dataset Erdős, which includes 50 graph-theoretic reasoning tasks from NetworkX, totaling 100k/2k training/evaluation question-answer pairs. Answers are automatically solved by NetworkX solvers. Graph data is downsampled from real-world datasets to 5-35 nodes, and graph structures are represented using a unified edge list format. The work then proposes applying a reinforcement learning (RL) framework to graph reasoning tasks for large language models (LLMs), designing rule-based rewards for different graph reasoning tasks and using Group Relative Policy Optimization (GRPO) for RL training. Finally, RL training on Erdős significantly improves the performance of baseline models on the Erdős benchmark.

**Questions:**

1)	My main concern is whether it is appropriate to use LLM to solve graph-theoretic problems without semantic information. Intuitively, the advantage of LLM lies in its ability to understand the textual attributes of nodes, so it is naturally suitable for scenarios such as text-attributed graphs. However, for pure graph structure problems addressed in the paper, isn't it more efficient to use pure GNN reasoning? On the other hand, tasks in Erdős all have clear optimization objectives, so wouldn't it be more efficient to solve them directly with algorithms? If the authors can provide sufficient reasons for using LLM, I am willing to raise my score.
2)	Corresponding to Question 1, it is hoped that the authors can supplement the experiments with comparisons of some other methods for solving graph-theoretic problems.
3)	Possibly due to limited space and rich content, Erdős is a newly proposed graph reasoning dataset by the authors, but it lacks analysis, such as whether it should be compared with existing other graph reasoning datasets.

**Ethical Concerns:**

["NO or VERY MINOR ethics concerns only"]

**Final Justification:**

Thanks for author's detailed responses, which addressed most of my concerns. I keep my positive score.

**Limitations:**

Yes

**Quality:**

4

**Strengths And Weaknesses:**

Strengths:
1)	Heavy workload. This work includes three main contributions: organizing the dataset, migrating the rule-based reward GRPO to the field of graph reasoning, and training the graph reasoning model.
2)	Novel theme. Improving the reasoning ability of LLMs on graphs is a relatively less explored field.

Weakness:

1)	The work mainly focuses on migrating GRPO to the graph reasoning field of large models, and the innovation of the work itself is relatively limited.
2)	There is a lack of comparison with other methods, such as directly using graph neural networks to solve graph reasoning problems, such as https://arxiv.org/abs/2209.11142
3)	The evaluation of the trained model is limited to Erdős, lacking assessment on other graph reasoning datasets.

---

> ### Author Rebuttal · Authors · 2025-07-31
>
> We thank Reviewer WDtd for recognizing our work's substantial contributions and for highlighting the novelty of our research theme in a less-explored field. We shall address your concerns point by point in the discussion below.
>
> ----
>
> **Q1.** My main concern is whether it is appropriate to use LLM to solve graph-theoretic problems without semantic information.
> > Intuitively, the advantage of LLM lies in its ability to understand the textual attributes of nodes, so it is naturally suitable for scenarios such as text-attributed graphs. However, for pure graph structure problems addressed in the paper, isn't it more efficient to use pure GNN reasoning [1]? On the other hand, tasks in Erdős all have clear optimization objectives, so wouldn't it be more efficient to solve them directly with algorithms? If the authors can provide sufficient reasons for using LLM, I am willing to raise my score.
>
> **A1.** Thanks for sharing your insightful thoughts! Indeed, for very standard, structural graph problems like those in Erdős, domain-specific solutions like GNNs are very compelling and efficient. In fact, the ultimate goal of this work is NOT to use LLMs to directly solve these problems, but by training on these graph-theoretic tasks, we want to improve **LLMs' general graph reasoning ability** for more real-world scenarios that involve graphs. In this way, we can leverage the power of LLMs to build ***general-purpose foundation models* capable of dealing with graphs.**
>
> **Why this is important** (and requires more than specialized GNNs) is that many real-world graph-related applications require sophisticated skills beyond domain-specific GNNs. For example, with a citation network, one may want to find out the top 10 influential people, and summarize their institutions in an Excel spreadsheet. This task is graph-related since the model needs to understand the graph well, but meanwhile, it's very hard for current GNNs that are only tasked to solve the graph part (and we need to train a model specifically for that particular task). With a graph-aware LLM, we can solve this with a simple instruction and release the potential of LLMs to expand their capability to more graph-related problems.
>
> In the paper, we benchmark this general graph understanding ability on common node/link prediction benchmarks to make it more standardized. We believe that this generic reasoning ability of G1 could unlock LLMs' potential for solving more graph-related tasks beyond standard ones. And training with synthetic graph-theoretic tasks is an effective recipe, as demonstrated by G1, to reliably enhance this capability. This is the vision that we share about this work and hope this clarify your question. Please let me know if there is more to clarify.
>
> **An Illustrative Example**
>
> Below, we use a concrete example in _GraphArena_ benchmark to illustrate the differences between graph-enhanced LLMs and GNNs. For this connected-components problem, a GNN-solver demands manual reformatting and coding of each variable, with delicate design for ambiguous descriptions like "representative node". Moreover, if the question shifts slightly —from "output one representative node from each component" to "output the component containing Angel Chavez"—the entire process must restart from scratch. By contrast, graph-enhanced LLMs (G1) unify these steps into a single question-answering framework, handling diverse problems without task-specific engineering while achieving strong performance (see Table 4). This flexibility, paired with LLMs' emergent reasoning, makes our direction promising for broader applications.
>
> *An illustrative example in GraphArena benchmark:*
> > You are required to identify all connected components in the given social network and output one representative node from each component.
> >
> > Problem to Solve: - Names in the network: Veronica Garcia, Katherine Brennan, Angel Chavez, Steven Martin, Brett Johnson, Megan Banks, Julia Dominguez, Rachel Mitchell - Fiendship connections: Veronica Garcia to Brett Johnson, Veronica Garcia to Megan Banks, Katherine Brennan to Brett Johnson, Katherine Brennan to Megan Banks, Angel Chavez to Megan Banks, Angel Chavez to Rachel Mitchell, Steven Martin to Megan Banks, Brett Johnson to Megan Banks, Megan Banks to Julia Dominguez, Megan Banks to Rachel Mitchell.
>
> [1]. Ibarz, Borja, et al. "A generalist neural algorithmic learner." LoG, 2022.
>
> [2]. Veličković, Petar, et al. "The clrs algorithmic reasoning benchmark." ICML, 2022.
>
> ---------------
>
>  **Q2.**  There is a lack of comparison with methods such as directly using GNNs to solve graph reasoning problems [1]. It is hoped that the authors can supplement the experiments with comparisons of some other methods for solving graph-theoretic problems.
>
> **A2.** Indeed, it would be more complete to include GNN results as well. Following your advice, we evaluate our models on the CLRS benchmark [1]. The benchmark only provides feature embeddings for the input, output, and hidden variables of each algorithm, which are specific for GNNs but not compatible with LLMs. Therefore, we sample a subset of the benchmark and translate feature embeddings into natural language descriptions. As seen in **Table R1**,  our G1-7B significantly outperforms GNN baselines on DFS (95.08% vs. 47.79%) and remains competitive on the shortest path task (98.19% vs. 85.71%). These results highlight G1-7B’s capability **as a general model** for solving graph-theoretic problems.
>
> *Table R1. Micro-F1 scores of G1-7B and GNN baselines on the CLRS benchmark (32 samples per task). GNN results are from [1].*
> | ~ | DFS |  DAG Shortest Paths |
> |---|-----|-----------|
> | Triplet-GMPNN [1] | _47.79_ | **98.19** |
> | Memnet [2] | 13.36 | 66.15 |
> | MPNN [3]   |  6.54 | 96.24 |
> | PGN [4]    |  8.71 | 96.94 |
> | Qwen2.5-7B-Instruct | 0.00 |  51.16  |
> | **G1-7B (Ours)** | **95.08** | _85.71_ |
>
> [1]. Ibarz, Borja, et al. "A generalist neural algorithmic learner." LoG, 2022.
>
> [2]. Sukhbaatar, Sainbayar, Jason Weston, and Rob Fergus. "End-to-end memory networks." NeurIPS, 2015.
>
> [3]. Gilmer, Justin, et al. "Neural message passing for quantum chemistry." ICML, 2017.
>
> [4]. Veličković, Petar, et al. "Pointer graph networks." NeurIPS, 2020.
>
> -----------
>
> **Q3.** The work mainly focuses on migrating GRPO to the graph reasoning field of large models, and the innovation of the work itself is relatively limited.
>
> **A3.** We understand the reviewer’s concern that we did not design a specific training objective (_e.g._, a GRPO variant) for the graph domain. However, this was intentional—we aimed to minimize graph-specific tweaks so that the G1 training recipe could easily align with the generic RL recipe for LLM training. Our results demonstrate a valuable insight:  **one does not need _extensive_ graph-specific tweaks to attain significant improvements of LLMs' graph reasoning ability**, which might be a surprise to years of graph research that emphasizes the need for domain-specific engineering on architectures or training objectives.
>
> Moreover, integrating graph reasoning into the current RL framework for LLMs is  **not a trivial task**. To ensure compatibility with our tasks, we carefully designed graph-specific rewards:
> - For tasks with unordered-set answers (_Section 4.1_), we replace strict value matching with **flexible set matching**, granting intermediate rewards for partially correct solutions.
> - For non-unique solutions, we implement **task-specific verification algorithms** to rigorously assess correctness.
> -  We also apply a  _soft_  data mixture strategy dynamically **reweighting rewards by task difficulty**, yielding better average performance than G1-3B ( _Appendix D_).
>
> These new technical designs systematically address graph reasoning’s unique challenges while remaining aligned with the generic RL framework. Thus, we believe that this approach might be a more generic and scalable way to build graph-aware foundation models.
>
> ---
>
> **Q4.** The evaluation of the trained model is limited to Erdős, lacking assessment on other graph reasoning datasets.
>
> **A4.** In **Section 5.2.1**, we have further evaluated G1 on two additional benchmarks - **GraphWiz** and **GraphArena** that have different graph structures, unseen reasoning tasks, and unfamiliar graph encoding formats. **Tables 3–4** show that G1 achieves strong zero-shot performance compared to other graph reasoning benchmarks, confirming its robustness to unseen graph types, tasks, and representations. This underscores our core claim: RL-incentivized reasoning generalizes beyond synthetic training environments.
>
> ---
>
> **Q5.** Possibly due to limited space and rich content, Erdős is a newly proposed graph reasoning dataset by the authors, but it lacks analysis, such as whether it should be compared with existing other graph reasoning datasets.
>
> **A5.**  Due to space constraints, we've included a comprehensive comparison of Erdős with existing graph reasoning benchmarks in **Appendix E.1**, covering: (1) task design and counts, (2) instance counts, (3) graph types, and (4) node sizes. Please let us know if you have some other aspects that might need further elaboration.
>
> -----
> Thanks for your comments and hope our answers could address your concerns. Please let us know if there is more to clarify. We are happy to address them during the discussion stage.

---

> > ### Comment · Reviewer_WDtd · 2025-08-05
> >
> > Thanks for author's detailed responses, which addressed most of my concerns. I keep my positive score.

---

### Decision · Program_Chairs · 2025-09-17

**Decision:**

Accept (poster)

**Comment:**

This work propose a novel method to integrate graph reasoning reward with GRPO-like RL based training schemes. While the authors use a rather heavy experiments workload to prove there point, the technical contribution and application is limited. To be specific, the practical concerns raised by some reviewer regarding the scalability of the proposed method is a major limitation. Since all of the reviewers agree on accepting this work, I will recommend to accep this work as a poster.